# Ovaries absent links dLsd1 to HP1a for local H3K4 demethylation required for heterochromatic gene silencing

Fu Yang[1]*, Zhenghui Quan[1], Huanwei Huang[1], Minghui He[1], Xicheng Liu[1], Tao Cai[1], Rongwen Xi[1,2]*

[1]National Institute of Biological Sciences, Beijing, China; [2]Tsinghua Institute of Multidisciplinary Biomedical Research, Tsinghua University, Beijing, China

**Abstract** Heterochromatin Protein 1 (HP1) is a conserved chromosomal protein in eukaryotic cells that has a major role in directing heterochromatin formation, a process that requires co-transcriptional gene silencing mediated by small RNAs and their associated argonaute proteins. Heterochromatin formation requires erasing the active epigenetic mark, such as H3K4me2, but the molecular link between HP1 and H3K4 demethylation remains unclear. In a fertility screen in female *Drosophila*, we identified *ovaries absent* (*ova*), which functions in the stem cell niche, downstream of Piwi, to support germline stem cell differentiation. Moreover, *ova* acts as a suppressor of position effect variegation, and is required for silencing telomeric transposons in the germline. Biochemically, Ova acts to link the H3K4 demethylase dLsd1 to HP1a for local histone modifications. Therefore, our study provides a molecular connection between HP1a and local H3K4 demethylation during HP1a-mediated gene silencing that is required for ovary development, transposon silencing, and heterochromatin formation.

**Editorial note:** This article has been through an editorial process in which the authors decide how to respond to the issues raised during peer review. The Reviewing Editor's assessment is that all the issues have been addressed (see decision letter).

DOI: https://doi.org/10.7554/eLife.40806.001

*For correspondence:
yangfu@nibs.ac.cn (FY);
xirongwen@nibs.ac.cn (RX)

Competing interests: The authors declare that no competing interests exist.

## Introduction

In eukaryotic genomes, heterochromatin is mainly composed of repetitive sequences such as transposons that require active silencing (*Slotkin and Martienssen, 2007*). Heterochromatin is defined by the presence of repressive epigenetic methylation of histone H3 at lysine 9 (H3K9me) and by heterochromatin protein 1 (HP1), which binds to H3K9me sites (*Lachner et al., 2001*; *Bannister et al., 2001*). Heterochromatin formation is mediated by co-transcriptional gene silencing, a process that requires small RNAs and their associated argonaute proteins (*Martienssen and Moazed, 2015*). In *Drosophila*, the argonaute protein Piwi and Piwi-interacting RNAs (piRNAs) use base-pairing to target nascent transcripts to the corresponding transposon regions. The Piwi/piRNAs then recruit gene silencing machinery, including HP1a and the H3K9 methyltransferase Egg to form heterochromatin (*Yang and Xi, 2017*; *Czech and Hannon, 2016*; *Brower-Toland et al., 2007*). The formation of heterochromatin also requires erasing of active epigenetic mark by the H3K4 demethylase dLsd1 (*Rudolph et al., 2007*), but the molecular link between HP1a and local H3k4 demethylation remains elusive.

Piwi, a founding member of the piRNA pathway in *Drosophila*, was initially identified as a fertility factor; its mutation results in germline degeneration and sterility (*Cox et al., 1998*; *Lin and Spradling, 1997*). To identify new genes involved in Piwi/piRNA-mediated gene silencing, we here conducted a female fertility screen by EMS mutagenesis and identified a novel recessive mutation on

**eLife digest** The complete set of genetic material within a cell is known as a genome. The genomes of human and other animal cells have regions of active genes interspersed with 'dark' regions known as heterochromatin, which contain genes and other types of genetic material that have been inactivated.

Heterochromatin commonly contains sections of genetic material known as transposons. When a transposon is active it is able to move around the genome, therefore, inactivating (or 'silencing') transposons helps to maintain the integrity of the genetic material in a cell. It is particularly important to silence transposons in the stem cells that produce sperm and egg cells – known as germline stem cells – to ensure genetic information is faithfully passed on to the next generation.

A protein called HP1a plays a major role in directing where heterochromatin forms in the genome. This process requires an enzyme called dLsd1 to remove a small tag from the genetic material but it is not clear how HP1a regulates the activity of dLsd1. To address this question, Yang et al. studied how egg cells form in fruit flies, which are often used as models of animal biology in experiments.

The team screened a population of fruit flies that carried mutations in many different genes to identify genes that affect the fertility of female flies. This revealed a gene named as *ovaries absent* (or *ova* for short) is required for egg cells to form. In germline stem cells *ova* silences transposons and in the surrounding tissue it represses a specific signal that usually maintains stem cells to allow the stem cells to divide to make egg cells. Further experiments using biochemical techniques found that the protein encoded by *ova* acts as a bridge to bring HP1a and dLsd1 together to silence genes in heterochromatin.

The next step would be to identify the functional counterpart of the *ova* gene in mammals, including humans, which may help to discover causes of infertility and develop new fertility treatment.

DOI: https://doi.org/10.7554/eLife.40806.002

the second chromosome. The homozygous mutant males are semi-lethal (*Supplementary file 1*, Table 1), but females are viable but do not lay any eggs; other than sterility, these females do not have other notable defects. Dissection revealed that these females had rudimentary ovaries: rather than a normal ovary, each oviduct in these mutant females was connected to only a tiny mass of cells (*Figure 1a,b*). Given this nearly 'ovaryless' phenotype, we named the gene associated with this mutation as *ovaries absent* (*ova*) and named this mutant allele *ova[1]* .

Complementation mapping with deficiency lines, followed by sequencing of candidate genes led us to identify a single nucleotide deletion in an exon of CG5694, which results in a truncated protein of 387 amino acids (aa) rather than the predicted 623 aa full length protein (*Figure 1—figure supplement 1a,b*). CG5694 encodes a protein with no obvious sequence similarity to any existing proteins in the NCBI database, but does have a conserved nuclear respiratory factor-1 (NRF-1)- like domain at its N-terminus (15–105 aa); this DNA-binding domain was initially identified in the mammalian transcription factor NRF-1 and are known to occur in at least one other *Drosophila* transcription factor, Erect Wing (Ewg) (*Figure 1—figure supplement 1c,d*). We used CRISPR-Cas9 to generate a knock-out allele in which the entire coding region of *CG5694* was deleted (*Figure 1—figure supplement 1e*). Homozygous knock-out allele females are sterile and exhibit virtually identical 'ovaryless' phenotypes as the *ova[1]* females (*Figure 1a*). Additionally, transgenic expression of a genomic DNA fragment containing the *ova* gene region was able to effectively rescue the ovary defect and restore fertility of *ova[1]* homozygous or *CG5694* null females (*Figure 1a,b*). Therefore, *ova* is allelic to *CG5694*.

Normally, oogenesis initiates in the germarium, an anterior part of the ovariole where germline stem cells (GSCs) reside. Each germarium normally harbors 2–3 GSCs that can be distinguished by spherically-shaped spectrosome and by their direct contact with the cap cell niche (*Figure 1c*). The decedents of GSCs move posteriorly as they differentiate into germline cyst, and then bud off from the germarium to form egg chambers (*Xie, 2013*; *Spradling, 1993*). Immunostaining of *ova[1]* homozygous and *ova[1/4]* trans-heterozygous ovaries revealed that the mutant ovaries completely lacked

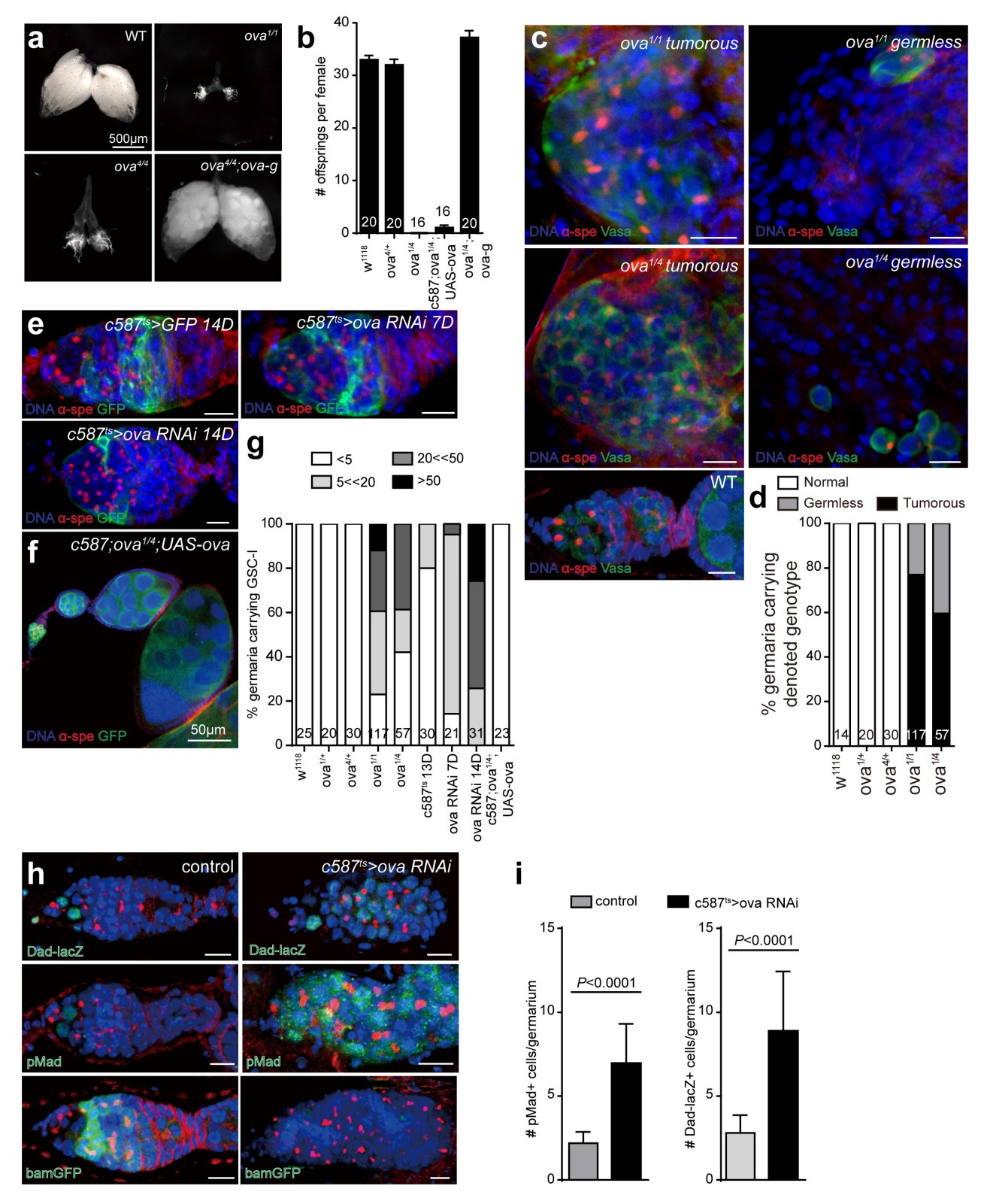

**Figure 1.** Ova is a niche factor for GSCs and ovary development in *Drosophila*. (a) Phase contrast images of dissected ovaries from flies of indicated genotypes. Scale bar, 500 μm. (b) A graph shows the total offspring number of indicated females (n = 20, 20, 16, 16, 20 respectively). (c) Representative image of germaria from indicated genotypes labeled by α-Spectrin (red), Vasa (green), and DAPI (blue). *ova*$^{1/1}$ and *ova*$^{1/4}$ ovaries have numerous spherical-shaped spectrosome-containing cells (tumorous) or are empty of germline cells (germless), indicated by lack of germline cell marker Vasa

*Figure 1 continued on next page*

*Figure 1 continued*

(green). A wild-type (WT) germarium is usually 2 GSCs localized to the anterior tip. Scale bar, 10 µm. (**d**) A graph shows the percentage of normal, germless, and tumorous germaria of indicated genotypes (n = 14, 20, 30, 117, 57 respectively). (**e**) c587$^{ts}$ > *ova* RNAi germarium accumulated GSC-like cells after shift to 29°C for 7 and 14 days. Scale bars, 10 µm. (**f**) Escort cell-specific expression of *ova* rescued oogenesis and GSC differentiation defect of *ova$^{1/4}$* females. Red, α-Spectrin; Green, Vasa (**g**) Quantification of GSC-like cell number in germaria of indicated genotypes (n = 25, 20, 30, 117, 57, 30, 21, 31, 23 respectively). (**h**) Confocal sections of germaria stained by indicated antibodies or reporter. Scale bars, 10 µm. (i) Quantitative results of pMad and Dad-lacZ positive cell numbers from germaria of indicated genotypes. Values are mean ± SEM.; n > 20. *P* values by two-tailed Student *t*-test.

DOI: https://doi.org/10.7554/eLife.40806.003

The following figure supplements are available for figure 1:

**Figure supplement 1.** Ova is allelic to CG5694.
DOI: https://doi.org/10.7554/eLife.40806.004

**Figure supplement 2.** Ova is not cell- autonomously required for GSC differentiation.
DOI: https://doi.org/10.7554/eLife.40806.005

vitellaria, and the germaria were either full of GSC-like cells [77% (n = 117) of *ova$^1$* germaria] (*Figure 1c,d,g*) or entirely germless (lacking Vasa expression (*Figure 1c,d,g*), suggesting that *ova* is required for GSC differentiation and for germline survival.

To determine whether *ova* functions cell-autonomously in the germline and/or non-cell-autonomously in somatic supporting cells to regulate GSCs, we conducted mosaic analysis by inducing mitotic clones using a FLP-FRT system (*Xu and Rubin, 1993*). Similar to wild-type control clones, *ova$^1$* mutant GSC clones behaved normally: the mutant GSCs were properly maintained in the niche, and their descendant cells were properly differentiated into germline cysts and egg chambers with properly specified oocytes (*Figure 1—figure supplement 2a–d*), although germline mutant eggs failed to hatch (*Figure 1b*). Similarly, germline-specific knocking down *ova* by UAS-Dcr2; nos-GAL4 (thereafter referred as *ova* GLKD) also did not cause any obvious defects in ovary morphology, and the number of GSCs and their immediate daughter cystoblasts (collectively referred to as GSC-like cells) per germarium remained largely normal (*Figure 1—figure supplement 2g,h*). Collectively, these data demonstrate that *ova* is not cell-autonomously required for the early stages of GSC differentiation. We next used a temperature sensitive GAL4/UAS system (*Brand and Perrimon, 1993*; *McGuire et al., 2004*) to specifically deplete *ova* in somatic escort cells with c587-GAL4 (c587 >ova RNAi) (*Song et al., 2004*). The somatic escort cells, which usually send out long protrusions that encapsulate the germline, are known to provide the niche environment required for germline cyst differentiation (*Kirilly et al., 2011*; *Morris and Spradling, 2011*). After treatment at the restrictive temperature, c587 >ova RNAi germaria began to exhibit a significantly increased number of spectrosome-containing GSC-like cells in a time-dependent manner (*Figure 1e,g*). The mutant escort cells were still able to send protrusions to the encapsulate the germline cells (*Figure 1e*), indicating that the GSC differentiation defects is likely not caused by defects in escort cell morphology. The requirement for *ova* i in somatic escort cells for proper GSC differentiation was further supported by the observation that escort cell-specific expression of an *ova* transgene was sufficient to rescue the ovary defects of *ova* mutant females (hereafter referred to as *ova* germline mutants) (*Figure 1f*). Therefore, *ova* functions in somatic escort cells and regulates germline differentiation in the germarium in a non-cell-autonomous manner.

The *ova* mutant phenotype is reminiscent of the *piwi* mutant phenotypes: *piwi* mutant flies also have rudimentary ovaries that contain an abnormal number of differentiation-blocked GSC-like cells, and *piwi* also functions primarily in somatic escort cells to regulate GSC differentiation (*Jin et al., 2013*; *Ma et al., 2014*). Loss of *piwi* in escort cells causes de-repression of *decapentaplegic* (*dpp*), a major self-renewal signal for GSCs, leading to GSC-like cell accumulation in the germarium (*Jin et al., 2013*; *Ma et al., 2014*). Interestingly, we found that the *ova* phenotype was also associated with increased *dpp* signaling. The extra GSC-like cells in c587 >ova RNAi germaria had dramatically increased expression of Dad-lacZ and phosphorylated Mad (pMad) (*Figure 1h,i*), both of which are reporters of BMP pathway activity, and decreased expression of *bam* (*Figure 1h*), a gene whose expression is normally suppressed by BMP signaling (*Song et al., 2004*; *Chen and McKearin, 2003*). Collectively, these data suggest that loss of *ova* in escort cells leads to ectopic *dpp* signaling that blocks further GSC differentiation, leading to GSC-like cell accumulation in the germarium.

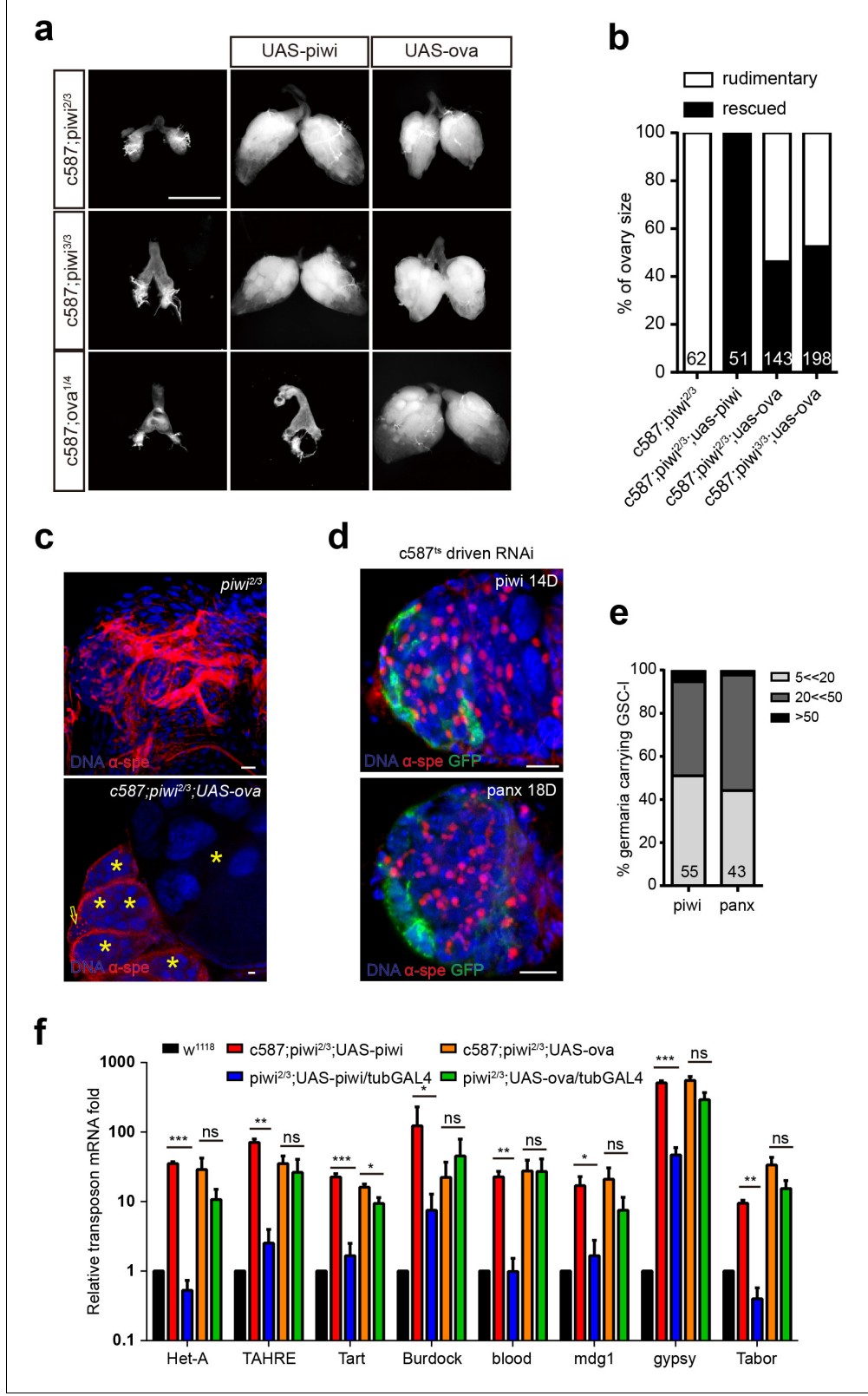

**Figure 2.** Ova acts downstream of Piwi genetically. (a) Phase contrast images of dissected ovaries from flies of indicated genotypes. Scale bar, 500 μm. (b) A graph shows the percentages of germaria with rudimentary or rescued ovaries (n = 62, 51, 143, 198 respectively). (c) Confocal sections of *piwi*[2/3] and *ova-rescued* ovaries. Arrow indicates the GSC-like tumor; asterisk indicates the developing germline cyst. Red, α-Spectrin. Scale bars, 10 μm. (d) Confocal sections of *piwi RNAi* and *panx RNAi* germaria. Red, α-Spectrin. Scale bars, 10 μm. (e) Quantification of GSC-like cell number in germaria of *Figure 2 continued on next page*

*Figure 2 continued*

indicated genotypes. (n = 55, 43 respectively). f, qPCR result of TE levels in total ovarian RNA from indicated genotypes (normalized to actin5c). Values are means ± SEM.; n = 3. *P* values by two-tailed *t*-test (*, p<0.05; **, p<0.01; ***, p<0.001).

DOI: https://doi.org/10.7554/eLife.40806.006

The phenotypic and molecular similarities between the *ova* and *piwi* mutants led us to further test whether Ova and Piwi act via the same genetic pathway to regulate GSCs. As expected, we observed that homozygous *piwi* mutant females had rudimentary ovaries. As a positive control, escort cell-specific expression of a *piwi* transgene was sufficient to rescue the *piwi* mutant ovary phenotype (*Figure 2a,b*). On the one hand, transgenic expression of *ova* in escort cells of *piwi* mutants also partially rescued the ovary morphology phenotype with the frequent appearance of developing germline cysts, including late stages of egg chambers, although oogenesis was still abnormal, GSC-like tumor still remained, and no mature eggs were produced (*Figure 2a,b,c*). On the other hand, transgenic expression of *piwi* in escort cells of *ova* mutants could not rescue any ovary phenotypes (*Figure 2a,b*). Consistent with previous reports of *piwi* phenotypes, escort cell-specific knocking down of other Piwi/piRNA pathway effectors, such as *panx*, also showed a similar GSC-l accumulation phenotype (*Figure 2d,e*), further supporting the idea that Ova may participate in the same Piwi/piRNA pathway. Next, we tested whether overexpression of *ova* could rescue the germline TE upregulation phenotype caused by *piwi* mutation. As a control, ubiquitous expression of *piwi*, but not soma-only expression of *piwi* was able to effectively rescue the TE upregulation phenotype. However, neither ubiquitous nor soma-specific expression of *ova* could rescue the TE upregulation phenotype in *piwi* mutants (*Figure 2f*). These observations suggest that, genetically, *ova* acts downstream of *piwi*, but there must be additional factors downstream of *piwi* that cooperatively function with *ova* to regulate GSC differentiation and transposon silencing.

Given that Piwi is associated with a number of chromatin factors that are known to regulate heterochromatin formation and germline transposon silencing, and considering that *dpp* silencing in escort cells can be attributed to Piwi-dependent gene silencing, we asked whether Ova is also associated with these silencing machinery components and somehow participates in these processes. We performed a yeast two-hybrid (Y2H) screen for potential physical interactions among Ova and other known essential components of the heterochromatin silencing machinery (*Yu et al., 2015*; *Sienski et al., 2015*; *Sienski et al., 2012*), including: Panoramix (Panx), Arx, and Mael, which participates in Piwi/piRNA-mediated gene silencing (*Yu et al., 2015*; *Sienski et al., 2015*; *Sienski et al., 2012*; *Muerdter et al., 2013*; *Dönertas et al., 2013*; *Ohtani et al., 2013*); HP1a, the H3K9me3 methyltransferase Eggless (Egg), and the Egg cofactor Windei (Wde) (*Seum et al., 2007*; *Tzeng et al., 2007*; *Koch et al., 2009*); the H3K4me2 demethylase dLsd1 and its cofactor CoREST (*Rudolph et al., 2007*); and Piwi. The Y2H screen identified two positive interactions: Ova and HP1a, and Ova and dLsd1 (*Figure 3a* and *Figure 3—figure supplement 1*). Notably, the previously reported interaction between HP1a and Piwi was not observed in our screen here (*Figure 3—figure supplement 1*) (*Brower-Toland et al., 2007*), possibly due to different expression systems used in the studies. Co-immunoprecipitation experiments also showed positive interactions between Ova and HP1a and between Ova and dLsd1 in ovary lysates (*Figure 3b,c*). Collectively, these results indicate that Ova is physically associated with the co-transcriptional silencing machinery and directly interacts with HP1a and with dLsd1. Previous studies have reported that HP1a and dLsd1 function in the escort cell niche to restrict *dpp* signaling and to facilitate GSC differentiation (*Wang et al., 2011*; *Eliazer et al., 2011*). These reports, considered alongside the known role of Piwi-dependent gene silencing of the *dpp* gene locus in normal escort cells, further supporting the notion that these three Piwi-associated factors (Ova, HP1a, dLsd1) function in a shared pathway in escort cells to establish a repressive chromatin state for the *dpp* gene locus.

We next tested whether Ova, similar to HP1a and dLsd1 (*Wang and Elgin, 2011*; *Czech et al., 2013*), is required for heterochromatin formation and germline transposon silencing. The *white* locus of In(l)w$^{m4h}$ chromosomal reversion flies is relocated to a position next to a heterochromatin region, and this relocation often causes heterochromatin-based silencing of this gene, resulting from a genomic phenomenon referred to as position effect variegation (PEV), these flies typically display mosaic eyes with red and white facets as a result of this relocation based silencing (*Wallrath and Elgin,*

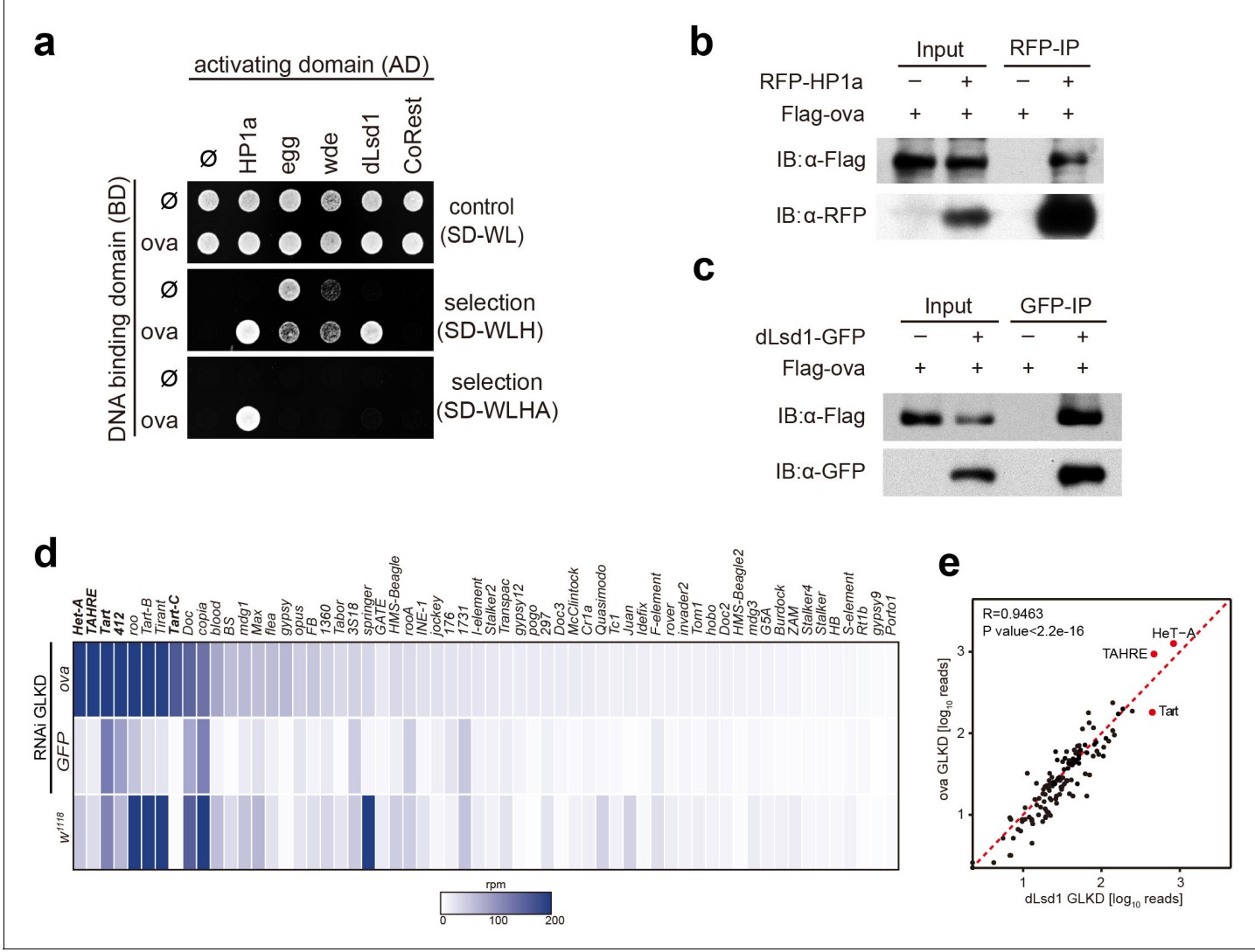

**Figure 3.** Ova interacts with the heterochromatin machinery. (a) Y2H assay for protein interaction between Ova and proteins as indicated. (b–c) Western blots showing reciprocal co-IP between Ova and HP1a, and between Ova and dLsd1. The RFP-HP1a transgene was driven by the endogenous promoter. The dLsd1-GFP transgene was driven by a ubiquitous promoter. The Flag-ova transgene was driven by nos-GAL4. (d) Heat map displaying steady state mRNA levels as reads per million (rpm) for the top 60 detected transposons in *nosGAL4* driven *ova-RNAi*, *EGFP-RNAi*, and *w1118* ovaries. The average of three replicates is shown. The most upregulated transposons are highlighted in bold. (e) Correlation scatter plot of $\log_{10}$ transposon mRNA-seq reads between *ova GLKD* and *dLsd1 GLKD* ovaries. R = 0.9463, p<$2.2\times10^{-16}$ by Pearson's correlation coefficient. The most upregulated transposons in both genotypes are highlighted in red dots.

DOI: https://doi.org/10.7554/eLife.40806.007

The following figure supplements are available for figure 3:

**Figure supplement 1.** Protein interaction mapping among components of heterochromatin machinery by Y2H assay.
DOI: https://doi.org/10.7554/eLife.40806.008

**Figure supplement 2.** Ova acts as a suppressor of position-effect variegation (PEV).
DOI: https://doi.org/10.7554/eLife.40806.009

**Figure supplement 3.** Ova is not required for piRNA biogenesis.
DOI: https://doi.org/10.7554/eLife.40806.010

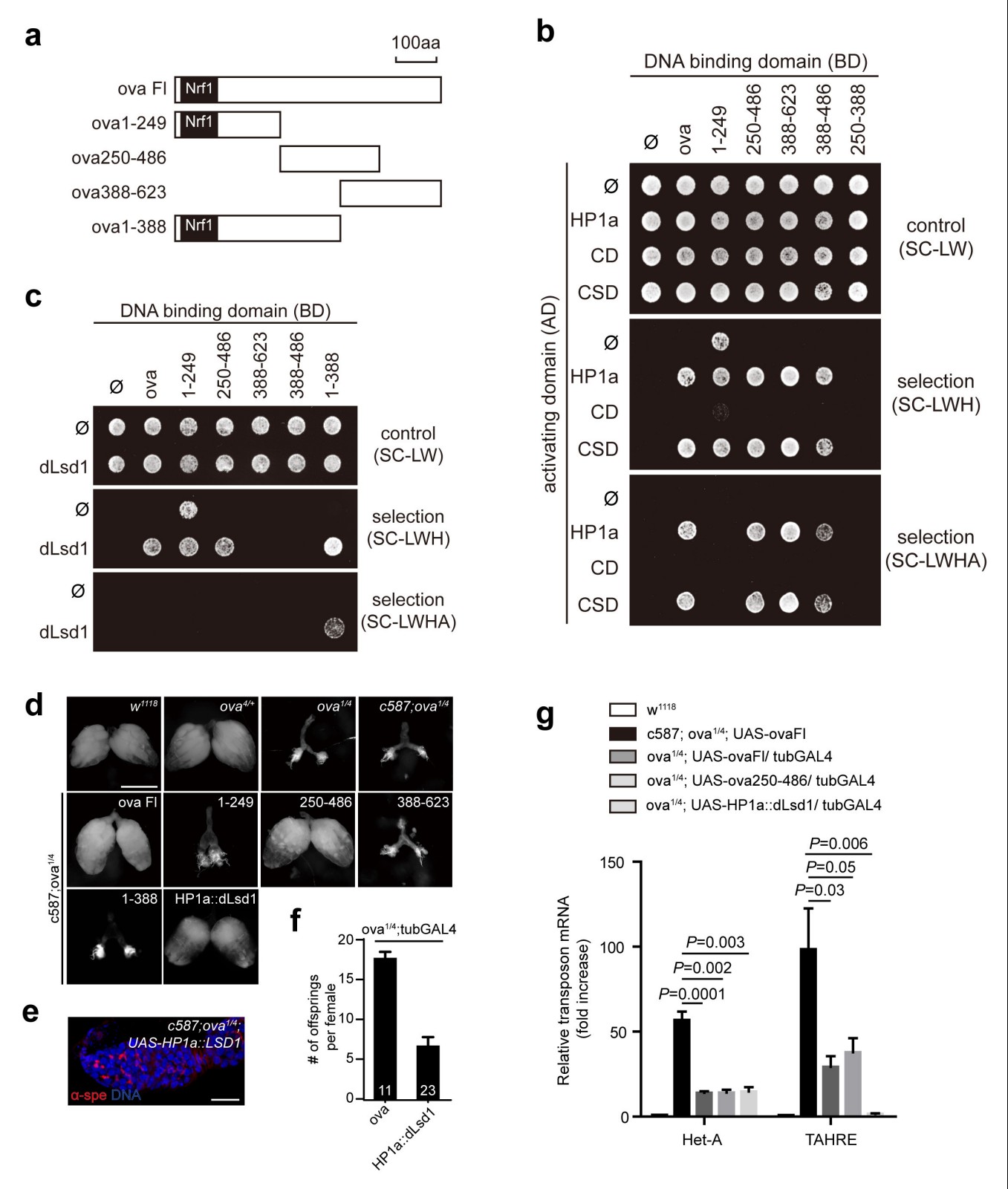

**Figure 4.** Ova acts as a protein adaptor to link dLsd1 with HP1a. (**a**) Schematic drawings of full length and truncated forms of Ova. (**b**) Mapping the reciprocal-binding regions between HP1a and Ova by Y2H assay. (**c**) Mapping the reciprocal-binding regions between HP1a and dLsd1 by Y2H assay. (**d**) Ovaries from flies of indicated genotypes. Escort cell-specific expression of ova full length, ova250-486 or HP1a::dLsd1 rescued *ova*^*1/4*^ ovary defect. Scale bar, 500 μm. (**e**) A representative image of *ova*^*1/4*^ germarium rescued by escort cell-specific expression of HP1a::dLsd1. Red, α-spectrin; Blue,

*Figure 4 continued on next page*

*Figure 4 continued*

DAPI. Scale bar, 10 μm. (**f**) A graph shows the total offspring number of indicated females (n = 11, 23 respectively). (**g**) A graph shows fold changes of TEs in total ovarian RNA from indicated genotypes (normalized to actin5c). Values are means ± SEM.; n > 4. *P* values by two-tailed *t*-test.

DOI: https://doi.org/10.7554/eLife.40806.011

The following figure supplement is available for figure 4:

**Figure supplement 1.** HP1a::dLsd1 expression rescues *ova* loss induced derepression of protein-coding genes.

DOI: https://doi.org/10.7554/eLife.40806.012

*1995*; *Schotta et al., 2003*) (*Figure 3—figure supplement 2a*). Interestingly, removing one functional copy of *ova* from the In(l)w^m4h background was sufficient cause fully-pigmented eyes (*Figure 3—figure supplement 2a*). Analysis using several additional PEV reporter fly lines (118E-10, 118E-15, 39 C-72, and 6 M-193), each of which has its *white* gene locus relocated (inserted) into the heterochromatin rich fourth chromosome, showed that *ova* acts as a suppressor of PEV: the *ova* transheterozygous flies had fully-pigmented eyes with increased pigment level whereas the *ova* heterozygous flies from all three of the reporter lines had mosaic eyes (*P* values by two-tailed Student *t*-test, *Figure 3—figure supplement 2b,c*). It thus appears that *ova* has a functional role in heterochromatic gene silencing.

To test whether or not *ova* functions in germline transposon silencing, we performed germline-specific knock-down of *ova* using the UAS-Dcr2; nos-GAL4 driver (*ova* GLKD), followed by RNA-seq analysis. Interestingly, nos > ova RNAi ovaries had dramatically up-regulated transcripts of a subset of transposons that included the LTR element 412 and the telomeric non-LTR repeats Het-A, TAHRE, Tart (*Figure 3d*). By comparison, the expression of protein-coding genes and piRNAs was largely un-altered (*Figure 3—figure supplement 3a,b,e*). Somatic cell-specific knock-down of *ova* (tj-GAL4 >ova RNAi) only caused mild, if any, TE upregulation (*Figure 3—figure supplement 3f*). Consistent with a role in germline transposon silencing, a previously reported genetic screen for genes involved in germline transposon silencing identified *ova* (CG5694) as one of the top hits (*Muerdter et al., 2013*; *Czech et al., 2013*). Notably, germline-specifc knock-down of either *ova* (*ova* GLKD) or *dLsd1* (*dLsd1* GLKD) exhibited de-repression of a similar subset of transposons (R = 0.9463 by Pearson's correlation coefficient, *Figure 3e*); this subset is distinguished by enrichment for bivalent histone marks (both H3K9me3 and H3K4me2) (*Czech et al., 2013*; *Klenov et al., 2014*). In mutants deficient in piRNA biogenesis, the inability to form Piwi/piRNA complexes typically results in retention of Piwi in the cytoplasm (*Wang and Elgin, 2011*; *Malone et al., 2009*; *Olivieri et al., 2010*). The fact that nuclear Piwi localization was largely unaffected in the *ova* mutant germline (*Figure 3—figure supplement 3c*) further supports our conclusion that *ova* is not required for piRNA biogenesis, but may function at the chromatin to mediate Piwi/piRNAs- induced transcriptional gene silencing, a phase that has been referred to as 'effector step' (*Czech et al., 2013*).

To explore the biochemical mechanisms underlying Ova function in greater detail, we used Y2H assays to identify the Ova domains required for its interactions with HP1a and/or dLsd1. We constructed multiple truncated forms of Ova (*Figure 4a*), and found that the Ova 250–486 fragment and the Ova 388–623 fragment were both able to interact with the chromo shadow domain (CSD) of HP1a (*Figure 4b*); neither of these Ova fragments could interact with the chromodomain (CD) of HP1a (*Figure 4b*). We next constructed an Ova fragment composed of the overlapped 388–486 region and confirmed that this fragment was sufficient for interaction with the CSD domain of HP1a (*Figure 4b*). Mapping the interaction domains of Ova with dLsd1 revealed that both Ova 1–388 and Ova 250–486 fragments, but not Ova 388–486 fragment, could interact with dLsd1 (*Figure 4c*). Interestingly, transgene expression of the Ova 250–486 fragment, which is able to interact with both HP1a and dLsd1, was sufficient to rescue both the ovary development defect and transposon silencing defect of *ova* mutant females, similar to the effect produced by transgene expression of a full length *ova* (*Figure 4d,f*). In contrast, no rescue effect was observed with the transgenic expression of the Ova 388–623 fragment, which interacts with HP1a only, or with expression of the Ova 1–388 fragment, which interacts with dLsd1 only (*Figure 4d*). Therefore, the domain that is sufficient to interact with both HP1a and dLsd1 is sufficient for Ova function in ovary development and transposon silencing. These biochemical and genetic experiments indicate that Ova may serve as a protein adaptor that links HP1a and dLsd1. To functionally test this putative adaptor function in vivo, we

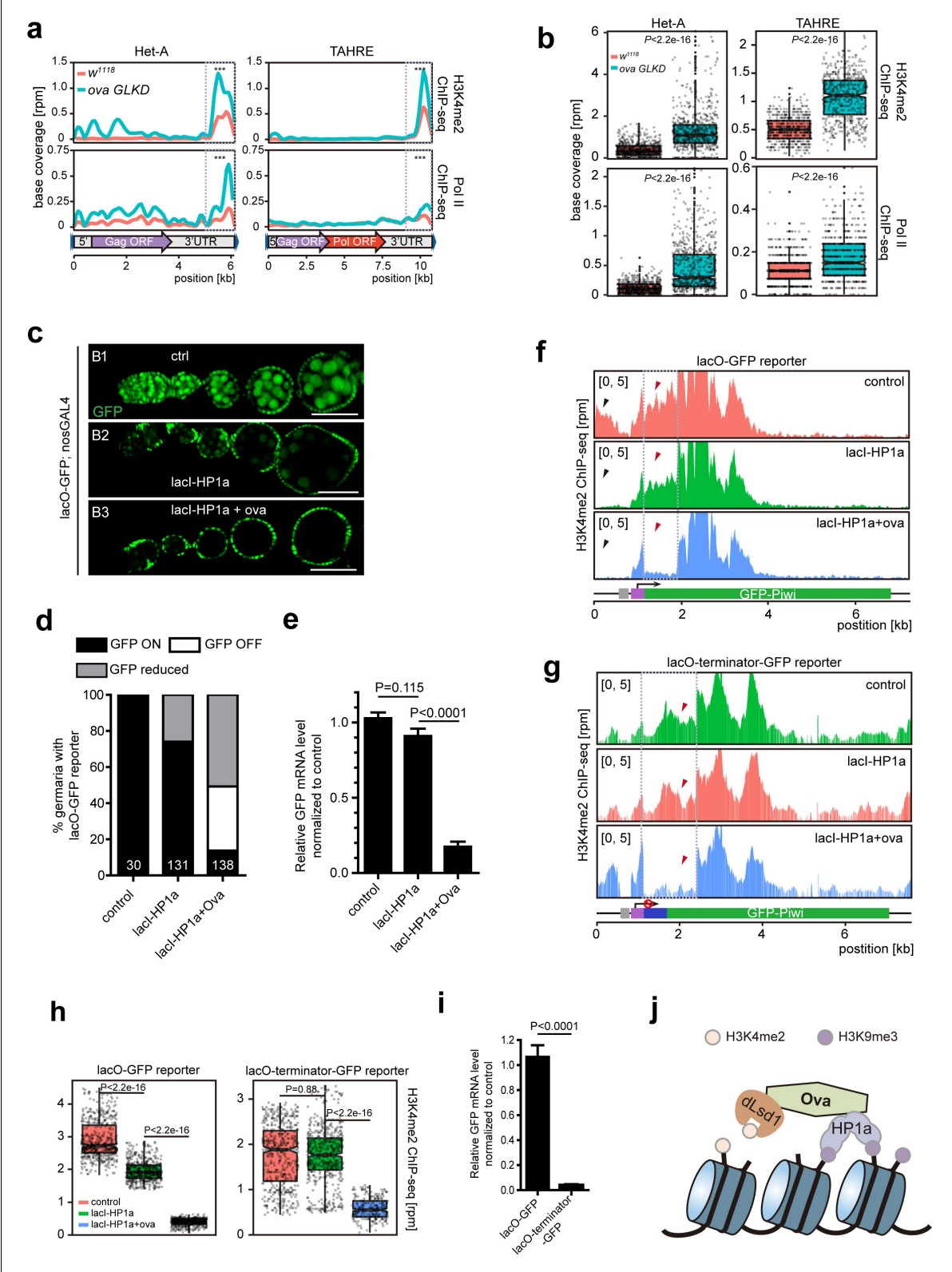

**Figure 5.** Ova regulates HP1a-induced local H3K4 demethylation. (a) Graphs showing H3K4me2 and Pol II ChIP-seq profiles mapped to indicated transposon loci in control versus ova GLKD ovaries. Dashed boxes, enhancer regions of transposons. RPM, reads per million. Bin, 100 bp. (b) Quantitative comparison of H3K4me2 and Pol II densities in the indicated enhancer regions in 5a (dashed boxes). *P* values by two-tailed Student's *t*-test. (c), Ovarioles with indicated genotypes expressing ubiquitous lacO-GFP reporter in the germline cells. GFP was visualized by antibody staining.

*Figure 5 continued on next page*

*Figure 5 continued*

Scale bar, 50 μm. (**d**) A graph showing the percentage of ovarioles of indicated genotypes with normal, reduced, or abolished GFP signals. (**e**) Quantitative RT-PCR results of GFP mRNA from ovaries of indicated genotypes. Values are means ±SEM.; n > 4. *P* values by two-tailed Student's *t*-test. (**f**) Graphs showing normalized H3K4me2 density mapped to lacO-GFP reporter region from ovaries of indicated genotypes. Grey box, lacO-binding sites; Purple box, nanos promoter. (**g**) Graphs showing normalized H3K4me2 density mapped to lacO-terminator-GFP reporter region from ovaries of indicated genotypes. Blue box, VASA terminator. (**h**) Quantitative comparison of H3K4me2 density in regions indicated by the dashed boxes in f or g. *P* values by two-tailed Student's *t*-test. (**i**) Quantitative RT-PCR results of GFP mRNA from lacO-GFP and lacO-terminator-GFP reporter ovaries. Values are means ± SEM.; n = 4. *P* value by Student *t*-test. (**j**) A schematic model for Ova function: Ova functions as a protein adaptor to link HP1a with dLsd1 for local H3K4 demethylation during HP1a-induced transcriptional gene silencing.

DOI: https://doi.org/10.7554/eLife.40806.013

The following figure supplements are available for figure 5:

**Figure supplement 1.** Comparison of H3K4me2 density on all transposons in w1119 and ova GLKD ovaries.

DOI: https://doi.org/10.7554/eLife.40806.014

**Figure supplement 2.** Tethering Ova to DNA or RNA is unable to induce co-transcriptional gene silencing.

DOI: https://doi.org/10.7554/eLife.40806.015

generated a transgene expressing HP1a::dLsd1 fusion protein. If Ova merely functions as an adapter that bridges the two proteins, the HP1a::dLsd1 transgene should render Ova dispensable and therefore should be able to rescue the *ova* mutant phenotypes. Strikingly, transgenic expression of HP1a::dLsd1 in escort cells was sufficient to rescue the rudimentary ovary phenotype of *ova* mutants (*Figure 4e*). Eighty percent of the HP1a::dLsd1 rescued germaria contained 2–5 GSC-l (n = 41) and all the germaria had properly differentiating cysts. Moreover, ubiquitous expression of HP1a::dLsd1 also significantly rescued the transposon silencing defects of *ova* mutants and partially restored female fertility (*Figure 4f*). Given that the genomic fragment transgene of *ova* (ova-g), which includes the cis-elements of *ova*, could fully restore fertility (*Figure 1a*), the incomplete rescue of fertility by the HP1a::dLsd1 fusion could be due to non-physiological levels of the transgene expression. Alternatively, *ova* could have additional roles beyond the adaptor role that are important for female fertility. In addition to increased expression of transposons, *ova* germline mutant ovaries also showed moderate upregulation of many protein-coding genes (*Figure 4—figure supplement 1*). Interestingly, this transgene expression also effectively brought the expression of many protein-coding genes back to wild-type levels (*Figure 4—figure supplement 1*). These observations indicate that HP1a and Ova may participate in transcriptional silencing of many regular protein-coding genes, in addition to transposons. We conclude that Ova acts as a protein adaptor to link HP1a and dLsd1 to promote HP1a-mediated gene silencing.

Since dLsd1 catalyzes H3K4me2 demethylation, Ova may function to link dLsd1 and HP1a for local H3K4 demethylation during heterochromatic gene silencing. Indeed, ChIP-seq analysis revealed that the H3K4me2 density was specifically increased at Het-A and TAHRE loci but not other TE loci (*Figure 5—figure supplement 1*). Further analysis revealed that there was a significant increase in H3K4me2 levels and in RNA Pol II occupancy at the 3'UTR of the Het-A and TAHRE transposons in *ova* GLKD ovarian germline cells (*Figure 5a,b*). Note that these telomeric transposons are arranged in a head-to-tail fashion; therefore, the 3' UTR of one element likely directs the transcription of its downstream neighbor (*Danilevskaya et al., 1997*). To further test this potential role of Ova in linking H3K4 demethylation during HP1a-mediated gene silencing in vivo, we used a clean lacI/lacO reporter system to tether lacI-HP1a to the promoter of a lacO-GFP reporter (*Sienski et al., 2015*). We found that 26% of ovarioles examined (n = 131) had reduced GFP signal in their germline upon lacI-HP1a induction (*Figure 5c,d*), although there was no significant reduced in the overall level of *GFP* mRNA (*P* value by two-tailed Student *t*-test, *Figure 5e*). Importantly, co-expression of *ova* in the germline caused a significant increase in the number of ovarioles with reduced or abolished GFP signal [86% (n = 138)], and the overall *GFP* mRNA level was also significantly reduced in these samples (*P* value by two-tailed Student *t*-test, *Figure 5c–e*). ChIP-seq analysis showed that this reduction in reporter expression was accompanied by significantly reduced H3K4m2 levels near the promoter region of the *GFP* gene reporter (*Figure 5f,h*).

To further confirm that the alteration of H3K4 deposition is a consequence of Ova recruitment, rather than a secondary effect following altered gene transcription, we performed a similar set of experiments, but with a modified lacO-terminator-GFP reporter that has a transcriptional terminator

immediately following the promoter (*Figure 5g*). This should result in blocked transcription no matter whether a transcriptional activator/repressor is present or not. As expected, this reporter showed a significant reduction of baseline transcription (down to approximately 3.8%) (*P* value by two-tailed Student *t*-test, *Figure 5i*). We found that tethering lacI-HP1a to the promoter failed to alter the H3K4me2 level proximal to the tethering site. Co-expression of Ova, however, almost erased entirely the H3K4me2 marks in the proximal region (*Figure 5g,h*). These observations further support the notion that Ova links HP1a and dLsd1 for local erasing of H3K4me2 marks.

We also tested whether Ova itself can induce gene silencing by tethering Ova directly to DNA and to mRNA using in vivo reporter systems in the ovarian germline. We used a lacI-Ova and lacO-GFP binary system to tether Ova to genomic DNA (*Sienski et al., 2015*) and found that such tethering did not have any obvious effect on GFP expression (*Figure 5—figure supplement 2a–c*). Similarly, tethering Ova to mRNA using a λN-Ova and GFP-boxB binary system did not cause any obvious effect on GFP expression (*Figure 5—figure supplement 2d–f*). These results are consistent with the idea that Ova acts downstream of HP1a in heterochromatic gene silencing.

Similar to other 'effector step' mutations, the loss of *ova* or *dlsd1* only causes de-repression of a subset of transposons; this is in contrast with the widespread transposon de-repression that is common in mutations affecting piRNA biogenesis (*Czech et al., 2013*). This disparity can possibly be explained by the existence of different silencing mechanisms for particular subsets of transposons. Illustrating this idea, our work suggests that transposons with bivalent histone marks may be preferential targets for Ova and dLsd1. A bivalent pattern of histone methylation may help to regulate the expression of transposons that require a delicate On/Off balance, for example with the expression of telomeric repeats known to be required for normal telomere function (e.g., Het-A, TAHRE, and Tart). The results of our study establishes that Ova has an indispensable role in facilitating dLsd1's H3K4 demethylation activity during HP1a-induced heterochromatic gene silencing and demonstrates that this Ova function is essential for germline development, heterochromatin formation, and Piwi/piRNA-mediated co-transcriptional gene silencing. Our study suggests that the Piwi/piRNA pathway may adapt a similar effector machinery to repress regular genes, such as the *dpp* gene in escort cells, in addition to TEs. A study in *S. pombe* reported a mechanism in which a RNAi protein complex links the activity of the H3K9 methyltransferase Clr4 with H3K4 demethylation by the H3K4 demethylase Lid2 (*Li et al., 2008*), indicating an evolutionarily conserved interplay of epigenetic marks during transcriptional gene silencing. Given that the mechanisms underlying heterochromatic gene silencing are known to be strongly conserved from *Drosophila* to mammals, an equivalent functional module that links HP1a with H3K4 demethylation likely exists in mammals as well.

## Materials and methods

**Key resources table**

| Reagent type (species) or resource | Designation | Source or reference | Identifiers | Additional information |
|---|---|---|---|---|
| Genetic reagent (Drosophila melanogaster) | ova[1] | This paper | | See Materials and methods |
| Genetic reagent (Drosophila melanogaster) | ova[4] | This paper | | See Materials and methods |
| Genetic reagent (Drosophila melanogaster) | c587-GAL4 | (*Song et al., 2004*) (DOI: 10.1242/dev.01026) | RRID:BDSC_67747 | |
| Genetic reagent (Drosophila melanogaster) | Dad-lacZ | (*Tsuneizumi et al., 1997*) (DOI: 10.1038/39362) | RRID:DGGR_118114 | |
| Genetic reagent (Drosophila melanogaster) | bam-GFP | (*Chen and McKearin, 2003*) | RRID:DGGR_118177 | |

*Continued on next page*

*Continued*

| Reagent type (species) or resource | Designation | Source or reference | Identifiers | Additional information |
|---|---|---|---|---|
| Genetic reagent (Drosophila melanogaster) | piwi[2] | (*Lin and Spradling, 1997*) | RRID:BDSC_43319 | |
| Genetic reagent (Drosophila melanogaster) | piwi[3] | (*Lin and Spradling, 1997*) | RRID:BDSC_12225 | |
| Genetic reagent (Drosophila melanogaster) | GFP-piwi | Katalin Toth (California Institute of Tchnology) | | |
| Genetic reagent (Drosophila melanogaster) | 118E-10 | Lori Wallrath (University of Iowa) | | |
| Genetic reagent (Drosophila melanogaster) | 118E-15 | Lori Wallrath (University of Iowa) | | |
| Genetic reagent (Drosophila melanogaster) | 6 M-193 | Lori Wallrath (University of Iowa) | | |
| Genetic reagent (Drosophila melanogaster) | 39C.72 | Lori Wallrath (University of Iowa) | | |
| Genetic reagent (Drosophila melanogaster) | dLsd1-GFP | Yu Yang (Institute of Biophysics, Chinese Academy of Science) | | |
| Genetic reagent (Drosophila melanogaster) | EGFP-RNAi | Bloomington Drosophila Stock Center | (#41553) | |
| Genetic reagent (Drosophila melanogaster) | RFP-HP1a | Bloomington Drosophila Stock Center | (#30562) | |
| Genetic reagent (Drosophila melanogaster) | UAS-Dcr2; nos-GAL4 | Bloomington Drosophila Stock Center | (#25751) | |
| Genetic reagent (Drosophila melanogaster) | tub-GAL4 | Bloomington Drosophila Stock Center | (#5138) | |
| Genetic reagent (Drosophila melanogaster) | tub-GAL80$^{ts}$ | Bloomington Drosophila Stock Center | (#7016, #7018) | |
| Genetic reagent (Drosophila melanogaster) | Df(2L)BSC144 | Bloomington Drosophila Stock Center | (#9504) | |
| Genetic reagent (Drosophila melanogaster) | attP2 | Bloomington Drosophila Stock Center | (#25710) | |
| Genetic reagent (Drosophila melanogaster) | In(1)w$^{m4h}$ | Kyoto Stock Center | (#101652) | |
| Genetic reagent (Drosophila melanogaster) | Df(2L)ED737 | Kyoto Stock Center | (#150520) | |
| Genetic reagent (Drosophila melanogaster) | ova-RNAi | Vienna Drosophila Research Center | (#102156) | |
| Genetic reagent (Drosophila melanogaster) | piwi-RNAi | Vienna Drosophila Research Center | (#101658) | |

*Continued on next page*

*Continued*

| Reagent type (species) or resource | Designation | Source or reference | Identifiers | Additional information |
|---|---|---|---|---|
| Genetic reagent (Drosophila melanogaster) | panx-RNAi | Vienna Drosophila Research Center | (#102702) | |
| Genetic reagent (Drosophila melanogaster) | EGFP-5xBoxB | Vienna Drosophila Research Center | (#313408) | |
| Genetic reagent (Drosophila melanogaster) | lacO-GFP-Piwi | Vienna Drosophila Research Center | (#313394) | |
| Genetic reagent (Drosophila melanogaster) | lacI-HP1a; lacO-GFP-Piwi | Vienna Drosophila Research Center | (#313409) | |
| Genetic reagent (Drosophila melanogaster) | nos-Cas9 | Jianquan Ni (Tsinghua University) | | |
| Genetic reagent (Drosophila melanogaster) | pCasper4-ova-g | This paper | | See Materials and methods |
| Genetic reagent (Drosophila melanogaster) | GFP-ova | This paper | | See Materials and methods |
| Genetic reagent (Drosophila melanogaster) | UASP-ova | This paper | | See Materials and methods |
| Genetic reagent (Drosophila melanogaster) | UASP-piwi | This paper | | See Materials and methods |
| Genetic reagent (Drosophila melanogaster) | UASP-ova1-249 | This paper | | See Materials and methods |
| Genetic reagent (Drosophila melanogaster) | UASP-ova250-486 | This paper | | See Materials and methods |
| Genetic reagent (Drosophila melanogaster) | UASP-ova388-623 | This paper | | See Materials and methods |
| Genetic reagent (Drosophila melanogaster) | UASP-ova1-388 | This paper | | See Materials and methods |
| Genetic reagent (Drosophila melanogaster) | UASP-HP1a::dLsd1 | This paper | | See Materials and methods |
| Genetic reagent (Drosophila melanogaster) | UASP-λN-ova | This paper | | See Materials and methods |
| Genetic reagent (Drosophila melanogaster) | UASP-lacI-ova | This paper | | See Materials and methods |
| Genetic reagent (Drosophila melanogaster) | lacO-terminator-GFP-Piwi | This paper | | See Materials and methods |
| Recombinant DNA reagent | UASP-λN | Julius Brennecke (Institute of Molecular Biotechnology) | | |
| Recombinant DNA reagent | UASP-lacI | | | |
| Recombinant DNA reagent | lacO-GFP-Piwi | | | |

*Continued on next page*

*Continued*

| Reagent type (species) or resource | Designation | Source or reference | Identifiers | Additional information |
|---|---|---|---|---|
| Recombinant DNA reagent | pGBKT7 | Clontech (Cat#630443) | | |
| Recombinant DNA reagent | pGAD | Clontech (Cat#630442) | | |
| Antibody | rabbit polyclonal anti-pMad | Ed Laufer (Columbia Universtity Medical Center) | RRID:AB_2617125 | IHC(1:1000) |
| Antibody | rabbit polyclonal anti-β-galactosidase | MP Biologicals (Cat#0855976) | RRID:AB_2687418 | IHC(1:3000) |
| Antibody | mouse monoclonal anti-α-Spectrin | Developmental Studies Hybridoma Bank | | IHC(1:50) |
| Antibody | mouse monoclonal anti-Tubulin | Developmental Studies Hybridoma Bank | RRID:AB_1157911 | WB(1:2000) |
| Antibody | rabbit polyclonal anti-mCherry | BioVision (cat#5993) | RRID:AB_1975001 | WB (1:2000) |
| Antibody | rabbit polyclonal anti-GFP | Life (cat#A11122) | RRID:AB_221569 | IHC(1:1000) WB(1:10000) |
| Antibody | polyclonal anti-rabbit IgG-HRP | ZSJQ-BIO (cat#ZB2301) | | WB(1:10000) |
| Antibody | rabbit polyclonal anti-H3K4me2 | Abcam (cat#ab7766) | RRID:AB_732924 | |
| Antibody | mouse monoclonal anti-RNA polymerase II | Abcam (cat#ab817) | RRID:AB_306327 | |
| Antibody | mouse monoclonal anti-Flag | Sigma (cat#F1804) | RRID:AB_439685 | IHC(1:300) WB(1:6000) |
| Chemical compound, drug | 4',6'-diamidino-2-phenylindole | Sigma (cat#10236276001) | | |
| Commercial assay or kit | anti-Flag resin | Sigma (cat#A2220) | RRID:AB_10063035 | |
| Commercial assay or kit | GFP-Trap agarose | Chromoteck (cat#gta-10) | | |
| Commercial assay or kit | RFP-Trap agarose | Chromoteck (cat#rta-10) | | |
| Commercial assay or kit | Qiagen Plasmid Midi Kit | Qiagen (#12145) | | |
| Commercial assay or kit | Immobilon Western Chemiluminescent HRP Substrate Kit | Millipore (cat#WBKLS0500) | | |
| Commercial assay or kit | HiScript II Q RT SuperMix | Vazyme Biotech (cat#R223-01) | | |
| Commercial assay or kit | ChamQ SYBR qPCR master Mix | Vazyme Biotech (cat#Q331) | | |
| Commercial assay or kit | Oligo d(T)$_{25}$ Magnetic beads | NEB (cat#S1419S) | | |
| Commercial assay or kit | NEBNext Ultra IIDNA Library Prep Kits for Illumina | NEB (cat# E7645S) | | |
| Commercial assay or kit | VAHTS Small RNA Library Prep Kit for Illumina | Vazyme Biotech (cat#NR801) | | |
| Commercial assay or kit | VAHTS Universal DNA Library Prep Kit | Vazyme Biotech (cat#ND607) | | |
| Commercial assay or kit | TruePrep Index Kit | Vazyme Biotech (cat#TD202) | | |

*Continued on next page*

*Continued*

| Reagent type (species) or resource | Designation | Source or reference | Identifiers | Additional information |
|---|---|---|---|---|
| Software, algorithm | GraphPad Prism | GraphPad Prism (https://graphpad.com) | RRID:SCR_002798 | |
| Sequenced-based reagent | RT-qPCR primers | This paper | | See *Supplementary file 1*, Table 2 |

### *Drosophila* strains

Flies were cultured on standard media with yeast paste added to the food surface. The culture temperature was 25°C unless otherwise noted. Strains used in this study were as follows: *ova[1]* is nucleotide loss allele (A1045) generated in this study. *ova[4]* is a knock-out allele generated in this study by CRISPR-Cas9 (*Ren et al., 2013*). *c587-GAL4* (*Song et al., 2004*); *Dad-lacZ* (*Tsuneizumi et al., 1997*); *bam-GFP* (*Chen and McKearin, 2003*); *piwi* (*Lachner et al., 2001*) and *piwi* (*Bannister et al., 2001*) (*Lin and Spradling, 1997*); *GFP-piwi* (gift from Katalin Toth, California Institute of Technology); *118E-10*, *118E-15*, *6* M-193, and *39C.72* (gift from Lori Wallrath, University of Iowa); *dLsd1-GFP* (gift from Yang Yu, Institute of Biophysics IBP, Chinese Academy of Sciences); from Bloomington *Drosophila* Stock Center (BDSC):*EGFP-RNAi* (#41553) *RFP-HP1a* (#30562);; *UAS-Dcr2*; *nos-GAL4* (#25751); *tub-GAL4* (#5138); *tub-GAL80[ts]* (#7016, #7018); *Df(2L)BSC144* (#9504); *attP2* (#25710); from Kyoto Stock Center: *In(1)w[m4h]* (#101652); *Df(2L)ED737* (#150520); from Vienna Drosophila Research Center: *ova-RNAi* (#102156); *piwi-RNAi* (#101658); *panx-RNAi* (#102702); *EGFP-5xBoxB* (#313408); *lacO-GFP-Piwi* (#313394); *lacI-HP1a*; *lacO-GFP-Piwi* (#313409).

### Generation of knock-out and transgenic flies

To obtain *ova* knock-out allele, two gRNAs (gRNA1: aagtctttacagccttgatc and gRNA2: cgttgggttgaggtacatac) were designed that target *ova* 5'UTR and 3'UTR respectively and cloned into U6b vector. The plasmids were introduced into *nos-Cas9* embryos (*Ren et al., 2013*). Obtained flies were backcrossed with *w[1118]* for at least three generations to eliminate potential off-target events. For *ova-g* transgenic fly, *w[1118]* genomic region (2L: 10226867–10234857) was cloned intro pCasper4 vector. The attP-UASP vector was used to generate *UASP-Flag-ova*, *UASP-Flag-ova1-388*, *UASP-Flag-ova1-249*, *UASP-Flag-ova250-486*, *UASP-Flag-ova-388–623*, *UASP-ova*, *UASP-piwi*, and *UASP-HP1a::dLsd1*. The *GFP-ova* construct was obtained using Gateway cloning technology (Invitrogen) and pUGW (DGRC1283) vector. Ova cDNA was cloned into UASP-λN and UASP-lacI (gifts from Julius Brennecke, Institute of Molecular Biotechnology) to generate the UASP-λN-ova and UASP-lacI-ova transgenes respectively. For the lacO-terminator-GFP reporter, 555 bp VASA terminator was injected immediately following start codon of GFP in the lacO-GFP reporter. All the plasmids were purified using a Qiagen Plasmid Midi Kit (#12145) and the DNA sequencing verified plasmids were introduced into embryos using either P-element or *nos-phiC31* system to generate transgenic flies according to a standard procedure.

### Immunostaining

*Drosophila* ovaries were dissected and immunostained as described previously (*Yang et al., 2015*). Briefly, ovaries were fixed in 4% paraformaldehyde for 15 min, and blocked in 5% normal goat serum in PBT (10 mM $NaH_2PO_4$, 175 mM NaCl, pH 7.4, 0.1% Triton X-100). The following primary antibodies were used: rabbit anti-pMad (1:1000, gift from Ed Laufer, Columbia University Medical Center, New York), rabbit anti-β-galactosidase (1:3000; MP Biologicals, 0855976), mouse anti-α-Spectrin (1:50; DSHB), rabbit anti-GFP (1:1000; Life, A11122), mouse anti-Flag (1:300; Sigma, F1804). Secondary antibodies, including goat anti-rabbit, anti-mouse IgGs, conjugated to Alexa (488 or 568) (Molecular Probes) were used at a dilution of 1:300 and tissues were also stained with 0.1 mg/ml DAPI (4',6'-diamidino-2-phenylindole; Sigma) for 5 min. Images were collected using either a Zeiss LSM510/LSM 800 or Nikon A1 confocal microscope system. All acquired images were processed in Adobe Photoshop and Illustrator.

## Fertility test

To test female fertility, for each vial, three newly enclosed females were collected and mated with three 5–7 days old $w^{1118}$ males in cornmeal food with yeast paste for two days, then the flies were transferred to a cornmeal food vial without yeast paste. After another three days, the flies were dumped out. The number of offspring was accounted until 16 days after eclosion. Mean values are reported as SEM.

## *Drosophila* eye pigmentation assay

To measure eye pigmentation, the heads of ten 5–7 days old flies of each genotype were manually dissected. The isolated heads were homogenized in 0.2 ml of methanol, acidified with 0.1% HCl and warmed at 50°C for 5 min; The homogenate was clarified by centrifugation, and the OD at 480 nm of 0.15 ml supernatant was recorded. Mean values are reported with SEM.

## Yeast two-hybrid experiment

Yeast Two-hybrid experiment was performed as described previously (*Yang et al., 2015*). Briefly, cDNA encoding interesting genes were amplified from $w^{1118}$ ovary cDNA and cloned into either pGBKT7 bait vector or pGAD prey vector (Clontech). The pGBKT7 and pGAD plasmid carrying interesting genes were co-transformed into AH109 yeast cells according to a standard procedure. Colonies appearing on media lacking tryptophan and leucine (SC-WL) were picked onto selection plate lacking tryptophan, leucine and histidine (SC-WLH) or tryptophan, leucine, histidine and adenine (SC-WLHA) to determine proteins interaction.

## Co-immunoprecipitation

Co-IP was done as previously described (*Yang et al., 2015*), with minor modifications. Female flies of appropriate genotypes were dissected in ice cold PBS. Ovaries were lysed in lysis buffer (10 mM Hepes pH 7.0, 150 mM NaCl, 5 mM $MgCl_2$, 10% glycerol, 1% Triton X-100, 1x complete protease inhibitor (Roche), 1 mM DTT, 1 mM EDTA, 0.1 mM PMSF) at 4°C for 30 min and spun for 10 min at max speed in a table top centrifuge at 4°C. The supernatant was incubated with tag-recognizing beads including anti-Flag resin (Sigma), GFP-Trap agarose beads (Chromoteck) and RFP-Trap agarose beads (Chromoteck). After incubation, the beads were washed three times with lysis buffer and eluted by boiling in SDS loading buffer, loaded onto SDS-PAGE gels, and analyzed by immunoblotting with indicated antibodies. The following primary antibodies were used: anti-Flag (Sigma, 1:6000), anti-GFP (Life, 1:10000), anti-mCherry (BioVision, 1:2000), anti-Tubulin (DSHB, 1:2000). Secondary antibodies, including: anti-mouse and anti-rabbit IgG-HRP (ZSJQ-BIO, 1:10000). The membrane was developed by Immobilon Western Chemiluminescent HRP Substrate Kit (Millipore) according to the manufacturer's instructions.

## RNA purification and real-time quantitative PCR (RT-qPCR)

Total RNA from 10 to 20 ovaries was extracted using TRIzol reagent (TaKaRa). After DNase treatment, complementary DNA (cDNA) was synthesized using HiScript II Q RT SuperMix (Vazyme Biotech, R223-01). RT-qPCR was performed in three duplicates using ChamQ SYBR qPCR master Mix (Vazyme Biotech, Q331) on an ABI PRISM 7500 fast real-time PCR system (Applied Biosystems). Fold changes for mRNA were calculated using the $\triangle\triangle$Ct method (*Livak and Schmittgen, 2001*). Primers used were shown in *Supplementary file 1*, Table 2.

## RNA sequencing and computational analysis

Total RNA from ovaries was isolated using TRIzol reagent (TaKaRa). 10 µg of total RNA from each sample used for library preparation after poly(A)-containing mRNA molecule purification (NEB, #S1419S), RNA amplification, double-strand cDNA synthesis, and adaptor ligation (NEB, #E7645S). For the small RNA sequencing, 10 µg enriched small RNA were separated on a 15% denaturing polyacrylamide gel and 18- to 30-nt RNAs were purified according to RNA oligo markers. All the libraries were prepared by following the manufacturer's instructions and subsequent sequencing on the Illumina GAII instrument (Vazyme, NR801). For CDS gene expression analysis, all the sequencing reads were mapped to the D.mel genome (BDGP6) using STAR program (options: –outFilterMultimapNmax 20 –alignIntronMin 20 –alignIntronMax 500000). The mapped reads were used for expression

analysis via Cufflinks package with reference gene annotation from Ensembl. And Cuffdiff was used to perform differential expression. For transposon expression analysis, sequencing reads were mapped to the transposon sequences which download from flybase website using STAR program with default parameters. Then alignment reads were used for calculating the expression level of transposons. Different transposons were combined together if they belong to the same one. The expression levels were normalized to reads per million (RPM). For small RNA analysis, Cutadapt package was used to remove adapter from 3' end. The reads were aligned to the genome sequence by Bowtie. The reads were discarded which mapped to rRNA, tRNA, snoRNA sequences. And retained reads were aligned to miRNA (pre-miRNA sequences download from miRBase) and whole genome sequences (r5.42) with one mismatch and unique hit. Sequences in the 25–32 nt size range, not annotated as a previously known RNA were classified as candidate piRNAs. The expression levels of small RNA were normalized to RPM according to the total mapped reads number.

## ChIP-seq analysis

ChIP was performed as previously described (*Sienski et al., 2012*). Briefly, about 200 pairs of ovaries were dissected into cold PBS and washed once. Ovaries were cross-linked in 1.8% paraformaldehyde for 10 min at room temperature then quenched with glycine. Ovaries were homogenized by douncing. Pellet was resuspended in lysis buffer and incubated 10 min on ice. Chromatin was sonicated for immunoprecipitation and followed by reverse crosslink and DNA purification. Recovered DNA fragment was used to prepare libraries using VAHTS Universal DNA Library Prep Kit (Vazyme, ND607) and TruePrep Index Kit (Vazyme Biotech, TD202) sequencing was done on HiSeq2500 (Illumina). Antibodies: polyclonal rabbit anti-H3K4me2 (Abcam, ab7766) and monoclonal mouse anti-RNA polymerase II (Abcam, ab817). ChIP-seq reads were aligned using Bowtie (version 1.1.2) to build version BDGP6 of the Drosophila melanogaster genome. MACS (version 1.4.1) was used to identify regions of ChIP-seq enrichment. The density of reads in each region was normalized to 10 million reads library size. For lacO-GFP ChIP-seq, normalized reads were removed $w^{1118}$ ChIP reads as the reporter unique mapped reads due to lacO-GFP reporter shared common sequences in fly genome. BigWig files were generated for visualization using Homer package. For transposons, all raw reads were mapped to the transposon database using Bowtie (version 1.1.2) with –v 3 –-best parameters. The sum of the number reads that mapped to genome and transposon was used as a normalization factor for all samples, reporting all feature abundances as RPM mapped.

## Acknowledgements

We thank the members of the fly community as cited in the Materials and Methods for generously providing fly stocks and antibodies, the Bloomington Drosophila Stock Center, Vienna Drosophila Stock Center, Tsinghua Fly Center, and Developmental Studies Hybridoma Bank (DSHB) for reagents, Dr. Bing Zhu for reading of the manuscript, and Drs. Julius Brennecke, Allan Spradling, and Yang Yu for scientific discussions. This work was supported by the National Key Research and Development Program of China (2017YFA0103602 to RX), National Basic Research Program of China (2014CB850002 to RX), and National Natural Science Foundation of China (31601059 to FY).

## Additional information

### Funding

| Funder | Grant reference number | Author |
| --- | --- | --- |
| Ministry of Science and Technology of the People's Republic of China | 2017YFA0103602, 2014CB850002 | Rongwen Xi |
| National Natural Science Foundation of China | 31601059 | Fu Yang |

The funders had no role in study design, data collection and interpretation, or the decision to submit the work for publication.

## Author contributions
Fu Yang, Conceptualization, Data curation, Formal analysis, Funding acquisition, Validation, Investigation, Visualization, Methodology, Writing—original draft, Writing—review and editing; Zhenghui Quan, Data curation, Formal analysis, Validation, Investigation, Visualization, Writing—original draft; Huanwei Huang, Data curation, Software, Formal analysis, Methodology; Minghui He, Data curation, Formal analysis, Investigation; Xicheng Liu, Data curation, Methodology; Tao Cai, Software, Formal analysis, Supervision; Rongwen Xi, Conceptualization, Resources, Formal analysis, Supervision, Funding acquisition, Investigation, Visualization, Writing—original draft, Project administration, Writing—review and editing

## Author ORCIDs
Xicheng Liu (iD) http://orcid.org/0000-0001-7257-1396
Rongwen Xi (iD) http://orcid.org/0000-0001-5543-1236

## Decision letter and Author response
Decision letter https://doi.org/10.7554/eLife.40806.021
Author response https://doi.org/10.7554/eLife.40806.022

## Additional files

### Supplementary files
• Supplementary file 1. Table 1: Viability test of *ova* mutants. Table 2: Primers used in this study.
DOI: https://doi.org/10.7554/eLife.40806.016
• Transparent reporting form
DOI: https://doi.org/10.7554/eLife.40806.017

### Data availability
High-throughput sequence data GEO link: https://www.ncbi.nlm.nih.gov/geo/query/acc.cgi?acc=GSE104925

The following dataset was generated:

| Author(s) | Year | Dataset title | Dataset URL | Database and Identifier |
|---|---|---|---|---|
| Huang H | 2018 | Impact of CG5694/ova in Drosophila ovaries | https://www.ncbi.nlm.nih.gov/geo/query/acc.cgi?acc=GSE104925 | NCBI , GSE104925 |

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
