## [Decision Letter]

[**Editorial note:** This article has been through an editorial process in which the authors decide how to respond to the issues raised during peer review. The Reviewing Editor's assessment is that all the issues have been addressed.]

Thank you for submitting your article "Ovaries absent links dLsd1 to HP1a for local H3K4 demethylation required for heterochromatic gene silencing" for consideration by *eLife*. Your article has been reviewed by three peer reviewers, and the evaluation has been overseen by a Reviewing Editor and Kevin Struhl as the Senior Editor. The following individuals involved in review of your submission have agreed to reveal their identity: Zhao Zhang (Reviewer #1); Felipe Karam Teixeira (Reviewer #2). Reviewer #3 remains anonymous.

The Reviewing Editor has highlighted the concerns that require revision and/or responses, and we have included the separate reviews below for your consideration. If you have any questions, please do not hesitate to contact us.

Summary:

In this manuscript, Yang et al. describes the role of new gene Ova required for female fertility. Whereas individual experiments are well-done overall, all reviewers are concerned that the results do not necessarily support a cohesive story at this point. Specifically, the effects in the soma vs. germline on overall sterility remains unclear, and accordingly, the impact/relevance of transposon activation in ova- on sterility is unclear. (The detailed, specific individual comments are provided below.)

Whereas all the reviewers appreciate the potential impact of this manuscript, these major concerns would be normally deemed 'not within the scope of straightforward revision' per *eLife*'s policy. Thus, we encourage the authors to thoroughly address the major points raised by reviewers in revising this manuscript.

Separate reviews (please respond to each point):

*Reviewer #1:*

In this manuscript, Yang and colleagues identified a new factor, Ovaries absent (Ova), which is required for maintaining the fertility of female fruit fly. Molecularly, this factor bridges H3K4 demethylase dLsd1 and HP1a together for heterochromatic silencing. At cellular level, the authors propose: 1) Ova functions in somatic escort cells to maintain the niche environment for germline stem cell differentiation by suppressing dpp signaling; 2) in germ cells, Ova promotes transposon silencing (only a subgroup of transposons) during oogenesis. The authors use a broad range of techniques (from genetics, biochemistry, cell biology), and the conclusions are overall convincing and interesting. I particularly think that the "rescuing" experiment by HP1a::dLsd1-fusion is elegant and compelling.

Here I suggest a few further experiments/explanations that can further strengthen this story:

1) I still do not understand in germ cells, how Ova is required for fertility. Is this exclusively caused by the activation of telomeric transposons? Or there are additional triggers? Since the HP1a::dLsd1 fusion can rescue the transposon de-repression, the authors should look and report the fertility of these "rescued" flies.

2) I feel the wm4h section is distracting and unnecessary in this story. Wm4h model is sensitive to genetic background, and likely reflects epigenetic regulation during eye development. I think it would be wise to focus on oogenesis for this story. With wm4h data, the readers (like me) would ask questions regarding how epigenetic marks maintained/passed through generations in somatic cells.

3) The qPCR results (panel T of Figure 1) on dpp are not conclusive and (again) unnecessary. The panels N-S of Figure 1 already provide compelling data at protein level in early germarium. I am assuming the qPCR data (panel T) are from the RNAs extracted from whole ovaries, which are mainly from the late stage egg chambers. Therefore, the RNA data (from late stage egg chambers) could not explain the change of protein level in germarium. Also, I am confused what is ova-RNAi in panel T. Is this c578 driven RNAi or nanos driven? The authors need to explain in figure legend.

4) The authors mentioned that Ova was identified from a genetic screen. But there is no information provided on this screen.

5) The ChIP-seq data currently only reported the K4me2 deposition on two transposons. How about the K4me2 level on other transposons and other genomic regions?

6) How about the HP1a and dLsd1 localization upon Ova depletion?

7) In Figure 4—figure supplement 1, what is the genotype of ova germline mutant (again, the figure legends are not informative, need to revise)? Are ovaries from these flies normal or rudimentary? If rudimentary, given the dramatic difference of ovary structure, it does not make sense to compare them with WT ovaries.

Minor Comments:

Typos in main text paragraph seven: "of in" and "Figure A".

Main text paragraph eleven: "80.43%". I think the number has to be written as words at the beginning of a sentence.

*Reviewer #2:*

In this manuscript, Yang and colleagues described a new factor required for adult ovary development in flies. Through an EMS screen, they uncovered a recessive mutation in the CG5694/ova gene, which led to drastic ovarian phenotype reminiscent of that observed in piwi mutants. Similar to Piwi, Ova is required in both ovarian somatic and germ cells, and somatic tissues depleted for Ova fail to support germline stem cell (GSC) differentiation. Importantly, the authors provided evidence that over-expression of Ova in ovarian somatic cells was able to partially rescue the phenotypic defects induced by piwi mutants, indicating that both genes genetically interact. By using two-hybrid experiments, the authors provided convincing evidence that Ova can physically interact with HP1a and Lsd1, two chromatin factors involved in establishing silence chromatin. In agreement, they showed that Ova is required for the spreading of heterochromatin in somatic tissues. Finally, by using transgenic reporters and tethering experiments, the authors provided evidence that Ova recruitment to an HP1a-bound reporter locus is sufficient to locally induce H3K4 demethylation – and activity that is mediated by Lsd1.

The antagonism between H3K4 and H3K9 methylation is pervasive throughout evolution, and the balance between activities promoting each of these marks is essential for proper genome regulation. The manuscript – which is clearly presented and well written – provides strong evidence that Ova can act as a linker between the H3K9me2/3 reader HP1a and the enzyme responsible for the removal of H3K4 methylation, proposing an interesting molecular mechanism for heterochromatic establishment.

Major comment:

Transposable elements are a major target of H3K9me2/3-mediated repression mechanisms in both soma and germline. However, in the current version, the experiment testing the effect of loss of ova on transposon repression is poorly documented (Figure 3E-F). Given the importance of such analysis for the conclusion of the manuscript, I believe this needs to be better characterized.

Basically, the evidence that Ova mediates transposon silencing is provided by germline KD experiments (Figure 3E), in which the nanos-GAL4 driver is used in combination with an UAS-RNAi line for the ova gene. RNA-seq results obtained from nanos-GAL4>ova-RNAi ovaries were compared to w[1118] ovaries, suggesting the upregulation of a limited number of TE families (basically the telomeric-associated Het-A, TAHRE, and TART families). First, it is well-established in the literature that germline KD using UAS-RNAi lines originated from the VDRC collection (such as the ova-RNAi line) requires the concomitant over-expression of Drc2 (using the UAS-Dcr2 transgene; Wang and Elgin, 2011; Handler et al., 2011; Czech et al., 2013; Handler et al., 2013; Yan et al., 2014; Sanchez et al., 2016; among others). Despite that, no mention to the use of the UAS-Drc2 transgene was seen in the main text or in the Materials and methods section – the authors need to clarify this point. Second, given the existing variation on transposon content in different stock backgrounds (especially true for the telomeric transposons), it is important to compare the results from ova-RNAi ovaries to appropriate control-RNAi samples (this is also valid for the lsd1-RNAi analysis). Finally, given the requirement of ova in somatic tissues, it would be important to determine whether the same specific regulation (restricted to telomeric-associated transposons) is observed in the soma (by using tj-Gal4 or C587-Gal4 drivers to induce somatic KD).

Along the same line, it would be very important to determine whether TE-family-specific changes in piRNA accumulation can be detected in Ova germline KD experiments (Figure 3—figure supplement 3B). As it is currently presented in Figure 3—figure supplement 3B, the results indicate that Ova is not required for global piRNA production, but it is nonetheless possible that it is involved in the accumulation of piRNAs for telomeric-associated transposons. Similar to the analysis presented in Figure 3E (RNA-seq), it would be good if the authors could expand the analysis on piRNA accumulation to each individual TE family (using the existing data).

Minor Comments:

1) While introducing the process leading to the Piwi/piRNA-mediated recruitment of the silencing machinery in the first paragraph of the main text, the authors mentioned that HP1 binds to Su(var)3-9. While it has been shown that HP1 and Su(var)3-9 physically interact and cooperate in the maintenance and spreading of heterochromatic domains (mostly centromeric heterochromatin) in somatic tissues (Schotta et al., 2002), there's no evidence to date that Su(var)3-9 is involved in Piwi-mediated silencing. To the contrary, Sienski et al., 2015, has recently provided substantial data indicating that SetDB1, but not Su(var)3-9, is involved in Piwi-mediated establishment of silent chromatin. In this context, I suggest the authors revise the text to make this distinction clear to readers.

2) Given the origin of the ova[1] allele (EMS-screen) and the rather strong and specific effect on male viability, it would be useful if the authors could include the viability data for ova[1]/ova[4] trans-heterozygous in Table S1. As it is, it is not clear whether the viability defect is due to ova or to other second-site mutations induced by the EMS-mutagenesis.

3) Main text paragraph eight, in the first sentence, it should be noticed that Piwi mediates transposon silencing in both soma and germline (and not only in the germline). Most important however, while it has been shown that loss of piwi in somatic cells is associated with ectopic dpp signalling (Jin et al., 2013; Ma et al., 2014), I am under the impression that the formal demonstration that this effect is direct – and that it involves gene silencing mediated by Piwi or piRNAs – is still missing. Could the authors clarify this point?

4) In paragraph ten, the authors state that "ova and lsd1 mutants exhibited de-repression of a similar subset of transposons" (Figure 3F). However, the analysis on Figure 3 seems to concern germline KDs, and not mutants – please clarify.

5) In the Materials and methods section, please verify the source of the ova-RNAi line: it is listed as a BDSC stock, but it seems like it was in fact obtained from the VDRC collection. Please also include the source of the lsd1-RNAi line, as it is not currently listed in the Materials and methods section. Finally, clarify whether or not the UAS-Dcr2 transgene was used in the germline KD analysis.

6) In subsection “RNA sequencing and computational analysis”, please indicate the manufacture and kits used to generate the small RNA- and RNA-seq libraries.

*Reviewer #3:*

In Yang et al., the authors generate mutants for ovaries absent (Ova) a previously identified gene in the piRNA pathway and characterize its function. They show (1) Ova is required in the somatic escort cells for germline stem cell differentiation and in the germ line for fertility, (2) Ova acts downstream of Piwi, (3) Interacts with HP1 and dLsd1 regulators of heterochromatin formation and promotes association of HP1a and dLsd1 and (4) Can induce HP1a induced demethylation. I have some major concerns about this paper: (1) The authors switch between germ line function and somatic function without explanation, (2) the phenotypes are not clearly characterized, (3) Some of the controls for the experiments they have carried out are not correct in that they compare ovaries that have late stage egg chambers to ovaries that accumulate undifferentiated cells and (4) changes in ChIPSeq are modest at the best and are not quantitated, and I am not sure are statistically significant.

Lastly, the model is confusing. The current dogma is that demethylation of H3K4 results loss of K9ac. This then promotes methylation of K9 and which then recruits HP1. In Yang et al's model, how do they propose HP1 can get to its targets in the first place? This needs to clearly explained. I suspect, Ova should affect heterochromatin spreading not formation of heterochromatin. This is something they have not tested nor propose.

Major concerns:

Figure 1A-D: The authors have used Ova1/4 in all the rescues including Figure 1 and 2. They should include the phenotypic characterization of this allelic combination in this figure panel.

Figure 1F-G: This panel needs better pictures as even the "germless" germaria usually stain of anti-spectrin antibody. All the pictures need to be at the same magnification for easy comparison.

Figure 1J-K: The controls should include the non-restrictive temperature and well as C587 at the restrictive temperature at 14 days. 14 days is a long time for restrictive temperature. What happens in 5 -7 days? Also, what happens to protrusions? The idea about protrusions were introduced but then not followed up on.

Figure 1M: Again, please include controls for ova4

Figure 1N-S: Please include quantitation for all these phenotypes. As they show a link to piwi and piwi mutant phenotype is quite complicated, they should categorize Ova phenotype in detail.

Figure 1T: They should not use C587 as a control as this is comparing completely morphologically different ovaries. One enriches for the earlier stages and the other for later stages. They should use bam mutants as a control. Additionally, they should show the RNA seq data and tracks for these experiments for Dpp levels – these data are available to them.

They also report that loss of ova in germ line results in eggs that do not hatch as data not shown. They should report the data.

Figure 2: While the results that over expression of ova in piwi mutant ovaries is exciting they should better characterize what aspect of the piwi mutant does this protein rescue? Are the transposons levels in piwi rescued? What about protrusions? What about Dpp levels? What about Ova germ line expression in piwi mutants? Does it rescue loss of fertility in piwi mutants as well? This is important as they switch from somatic effects to germ line effects.

Figure 3: What happens to heterochromatin levels in ova mutants during oogenesis? Can they stain for H3K9me3 marks?

Figure 3E: Why are the authors looking for only upregulation of transposons in the germ line when the phenotype is mostly from the soma? They should report transposon levels in somatic loss of function of ova. As the major phenotype of piwi, dlsd1 and ova are from the escort cells. They should carry out an RNAseq for this and report the transposons upregulated in the soma.

Figure 4: In this figure, the authors switch back and forth between the somatic effects and germ line. This is not right. They should be direct that for expediency they use germ line as a read out. They should set up the paper that way as well. The paper reads as if one is discovering the role of Ova in escort cells whereas most of the functional data comes from the germ line. For example 4D vs 4F, in D they are using somatic drivers and in F they use a ubiquitous driver and measure germ line effects.

Figure 4F: Again, using *w1118* as control is not right. One does not know if the undifferentiated stages express Het-A and TAHRE at higher levels compared to control. Again, the control and experiment are comparing two different morphologically different ovaries.

Figure 5A: Is this germ line or somatic KD? The authors should clearly state this. It is not available in text of figure legends. I assume it is germ line KD. How does this change compare with the genes that are not affected? Is there any quantification for these changes? I am not sure these changes 0-0.75 are real?

There should be quantification for panels A, E and F.

---

## [Author Response]

Whereas all the reviewers appreciate the potential impact of this manuscript, these major concerns would be normally deemed 'not within the scope of straightforward revision' per eLife's policy. Thus, we encourage the authors to thoroughly address the major points raised by reviewers in revising this manuscript.

First, we would like to thank all of the reviewers for their careful reading of our manuscript and for their insightful and constructive comments and suggestions. In this revised manuscript, we have made intensive efforts to address each of the comments. Kindly note, however, that we have not yet finished the ChIP-seq analysis for H3K9me owing to ongoing technical challenges that we have encountered. Other than that, we feel confident that the reviewers will find we have adequately addressed the rest of the comments. As we trust you'll agree, both our study and the manuscript have been significantly improved in the course of our revision process. We believe that the current version should now be suitable for publication in *eLife*. Again, many thanks for your ongoing work on our behalf.

Separate reviews (please respond to each point):

Reviewer #1:

[…] Here I suggest a few further experiments/explanations that can further strengthen this story:1) I still do not understand in germ cells, how Ova is required for fertility. Is this exclusively caused by the activation of telomeric transposons? Or there are additional triggers? Since the HP1a::dLsd1 fusion can rescue the transposon de-repression, the authors should look and report the fertility of these "rescued" flies.

We have conducted the fertility test of these “rescued” flies, and the results have been included in updated Figure 4F. Female flies of *ova^1^/ova^4^* are completely sterile (Figure 1B). As a positive control, ubiquitous overexpression of Ova with tubGAL4 could restore approximately 50% fertility of *ova^1^/ova^4^* females compared to WT and heterozygous controls (Figure 1B). Ubiquitous overexpression of HP1a::dLsd1 fusion protein could restore approximately 20% fertility, suggesting that this adaptor role of Ova is important for female fertility. Given that the genomic fragment transgene of *ova (ova-g*), which includes the cis-elements of *ova*, could fully restore fertility, the incomplete rescue of fertility by the HP1a::dLsd1 fusion could be due non-physiological levels of the transgene expression. Alternatively, *ova* could have additional roles beyond the adaptor role that are important for female fertility.

2) I feel the wm4h section is distracting and unnecessary in this story. Wm4h model is sensitive to genetic background, and likely reflects epigenetic regulation during eye development. I think it would be wise to focus on oogenesis for this story. With wm4h data, the readers (like me) would ask questions regarding how epigenetic marks maintained/passed through generations in somatic cells.

We appreciate the reviewer’s concern about this Wm4h model, so we have moved the PEV results to the Figure 3—figure supplement 2. Please note that we have tested Wm4h assay in at least two different genetic backgrounds: *ova^1^* and *ova^4^* heterozygous males, and they all show similar results. The effect on PEV is also found in several other PEV models in which the white gene is inserted on the fourth chromosome (Figure 3—figure supplement 2). It is thus clear that Ova is a bona fide suppressor of PEV. We think this could be a useful piece of data for the understanding of *ova* function, and have therefore decided to include them in the supplement of this manuscript.

3) The qPCR results (panel T of Figure 1) on dpp are not conclusive and (again) unnecessary. The panels N-S of Figure 1 already provide compelling data at protein level in early germarium. I am assuming the qPCR data (panel T) are from the RNAs extracted from whole ovaries, which are mainly from the late stage egg chambers. Therefore, the RNA data (from late stage egg chambers) could not explain the change of protein level in germarium. Also, I am confused what is ova-RNAi in panel T. Is this c578 driven RNAi or nanos driven? The authors need to explain in figure legend.

We agreed that there is abundant *dpp* expression in late stage egg chambers, which makes the qPCR analysis unnecessary. We have removed the data from the Figure 1. The ova-RNAi in panel T is driven by the c587-GAL4 driver, so *ova* was specifically knocked down in escort cells. We have included the driver information in the corresponding figures and legends throughout the manuscript to help avoid confusion.

4) The authors mentioned that Ova was identified from a genetic screen. But there is no information provided on this screen.

We have generated about 3000 EMS mutagenized lines for the FRT42B chromosome (the right arm of the second chromosome) to be used for other purposes (mosaic genetic screens). About 500 lines from this collection were homozygous viable, and were subsequently tested for female fertility. The *ova^1^*line was initially identified from this small-scale fertility screen. The FRT42B site in the *ova^1^* was later removed by meiotic recombination. Through deficiency map and other genetic analyses as described in the paper, we found that *ova^1^* is allelic to CG5694, which is actually localized on the left arm of the second chromosome. We used a standard EMS mutagenesis protocol, and will describe the details elsewhere.

5) The ChIP-seq data currently only reported the K4me2 deposition on two transposons. How about the K4me2 level on other transposons and other genomic regions?

We have re-analyzed the ChIP-seq data on all the transposons (see the newly added Figure 4—figure supplement 1). H3K4me2 density is increased in Het-A and TAHRE loci, but not in the majority of other transposons. This result is consistent with the idea that Ova preferably regulates transposons at the telomeric transposons by regulating H3K4m2 levels.

6) How about the HP1a and dLsd1 localization upon Ova depletion?

We have done immunostaining against HP1a and dLsd1 in control and ova GLKD ovaries, and the results are shown in Author response image 1. There is no obvious difference in terms of general expression levels and subcellular localization between wild type and *ova* mutant ovaries. As HP1a and dLsd1 have thousands of genomic target sites, the more precise ChIP-seq experiments, which are technically challenging, need to be done in the future to address this question.

**Author response image 1. respfig1:** dLsd1 and HP1a expression and subcellular localization in control and ova GLKD germaria

7) In Figure 4—figure supplement 1, what is the genotype of ova germline mutant (again, the figure legends are not informative, need to revise)? Are ovaries from these flies normal or rudimentary? If rudimentary, given the dramatic difference of ovary structure, it does not make sense to compare them with WT ovaries.

Thanks for catching this point; we indeed did not describe the genotypes clearly, which likely caused unnecessary confusion. The genotype of the germline mutant flies in that figure is *c587/+; ova^1^/ova^4^; UAS-ova/+*, and the ovaries from these mutant females are morphologically normal (see Figure 4D). Please note that the germline-specific *ova* mutants have morphologically normal ovaries, and this is also true for *ova* GLKD mutants (ova was specifically knocked down in the germline, see the updated Figure 1—figure supplement 2G). We have included the genotypes in the corresponding figure legends

Minor Comments:Typos in main text paragraph seven: "of in" and "Figure A".

Corrected.

Main text paragraph eleven: "80.43%". I think the number has to be written as words at the beginning of a sentence.

Corrected.

Reviewer #2:

[…] The antagonism between H3K4 and H3K9 methylation is pervasive throughout evolution, and the balance between activities promoting each of these marks is essential for proper genome regulation. The manuscript – which is clearly presented and well written – provides strong evidence that Ova can act as a linker between the H3K9me2/3 reader HP1a and the enzyme responsible for the removal of H3K4 methylation, proposing an interesting molecular mechanism for heterochromatic establishment.Major comment:Transposable elements are a major target of H3K9me2/3-mediated repression mechanisms in both soma and germline. However, in the current version, the experiment testing the effect of loss of ova on transposon repression is poorly documented (Figure 3E-F). Given the importance of such analysis for the conclusion of the manuscript, I believe this needs to be better characterized.

This is an excellent point. We have tried several times for ChIP-seq analysis of H3K9me2/3 from wild type and germline mutant ovaries, but unfortunately we failed to get any high quality data. This part of the analysis has therefore been hampered by the technical difficulties that we have encountered. We immunostained H3K9me3 in ovary tissues to see if there is a global change in H3K9me3 levels, and we failed to observe any obvious alterations in H3K9me3 expression in the germline cells with *ova* GLKD.

Basically, the evidence that Ova mediates transposon silencing is provided by germline KD experiments (Figure 3E), in which the nanos-GAL4 driver is used in combination with an UAS-RNAi line for the ova gene. RNA-seq results obtained from nanos-GAL4>ova-RNAi ovaries were compared to w[1118] ovaries, suggesting the upregulation of a limited number of TE families (basically the telomeric-associated Het-A, TAHRE, and TART families). First, it is well-established in the literature that germline KD using UAS-RNAi lines originated from the VDRC collection (such as the ova-RNAi line) requires the concomitant over-expression of Drc2 (using the UAS-Dcr2 transgene; Wang and Elgin, 2011; Handler et al., 2011; Czech et al., 2013; Handler et al., 2013; Yan et al., 2014; Sanchez et al., 2016; among others). Despite that, no mention to the use of the UAS-Drc2 transgene was seen in the main text or in the Materials and methods section – the authors need to clarify this point. Second, given the existing variation on transposon content in different stock backgrounds (especially true for the telomeric transposons), it is important to compare the results from ova-RNAi ovaries to appropriate control-RNAi samples (this is also valid for the lsd1-RNAi analysis). Finally, given the requirement of ova in somatic tissues, it would be important to determine whether the same specific regulation (restricted to telomeric-associated transposons) is observed in the soma (by using tj-Gal4 or C587-Gal4 drivers to induce somatic KD).

We appreciate the reviewer’s points and apologize for not clearly describing our methods. We indeed used the BSC#25751 fly, whose genotype is UAS-Dcr2;nosGAL4 (referred as "GLKD" throughout the updated manuscript). We compared the *w1118, ova RNAi* alone, and UAS-Dcr2;nosGAL4 ovaries by qPCR and found no obvious difference among them, as shown in Author response image 2. We used tjGAL4 to knock down ova and found that TE mRNA levels only mildly increased (not statistically significant) compared to the controls, as shown in Figure 3—figure supplement 3F. It is possible that the upregulation has been underestimated because we used the whole ovary for RNA analysis, but it appears to us that the transposon derepression phenotype is much more profound in the germline than in the soma following ova depletion.

**Author response image 2. respfig2:** qPCR to detect TE levels in controls and *ova* GLKD ovaries.

Along the same line, it would be very important to determine whether TE-family-specific changes in piRNA accumulation can be detected in Ova germline KD experiments (Figure 3—figure supplement 3B). As it is currently presented in Figure 3—figure supplement 3B, the results indicate that Ova is not required for global piRNA production, but it is nonetheless possible that it is involved in the accumulation of piRNAs for telomeric-associated transposons. Similar to the analysis presented in Figure 3E (RNA-seq), it would be good if the authors could expand the analysis on piRNA accumulation to each individual TE family (using the existing data).

Good point. Guided by this suggestion, we performed the analysis of the piRNA accumulation in each TE-specific piRNAs, and found that most TE-specific piRNAs are also unchanged in *ova* RNAi ovaries compared to control ones, excepting a mild reduction in Tart-specific piRNAs. This new data is included in the updated Figure 3—figure supplement 3E.

Minor Comments:1) While introducing the process leading to the Piwi/piRNA-mediated recruitment of the silencing machinery in the first paragraph of the main text, the authors mentioned that HP1 binds to Su(var)3-9. While it has been shown that HP1 and Su(var)3-9 physically interact and cooperate in the maintenance and spreading of heterochromatic domains (mostly centromeric heterochromatin) in somatic tissues (Schotta et al., 2002), there's no evidence to date that Su(var)3-9 is involved in Piwi-mediated silencing. To the contrary, Sienski et al., 2015, has recently provided substantial data indicating that SetDB1, but not Su(var)3-9, is involved in Piwi-mediated establishment of silent chromatin. In this context, I suggest the authors revise the text to make this distinction clear to readers.

Thanks for this input. We have revised the introduction accordingly.

2) Given the origin of the ova[1] allele (EMS-screen) and the rather strong and specific effect on male viability, it would be useful if the authors could include the viability data for ova[1]/ova[4] trans-heterozygous in Table S1. As it is, it is not clear whether the viability defect is due to ova or to other second-site mutations induced by the EMS-mutagenesis.

We have included the *ova[1]/ova[4]* viability in updated Table S1. Similar to *ova[1]* homozygous flies, *ova[1]/ova[4]* is semi-lethal for male.

3) Main text paragraph eight, in the first sentence, it should be noticed that Piwi mediates transposon silencing in both soma and germline (and not only in the germline). Most important however, while it has been shown that loss of piwi in somatic cells is associated with ectopic dpp signalling (Jin et al., 2013; Ma et al., 2014), I am under the impression that the formal demonstration that this effect is direct – and that it involves gene silencing mediated by Piwi or piRNAs – is still missing. Could the authors clarify this point?

Great point. Consistent with previous reports, we have knocked down two well-known piRNA pathway genes, *piwi* and *panx*, in escort cells and found that they have similar phenotypes with *ova* (Figure 2D, E). Together with the finding that HP1a::dLsd1 fusion protein could render *ova* dispensable for germline differentiation, we believe that the Piwi pathway adapts a similar effector machinery to repress *dpp* and TEs in escort cells. We agree that although many observations are consistent with the idea that *dpp* in escort cells is silenced by piRNA-mediated gene silencing machinery, the direct evidence, such as direct binding of Piwi or Ova to the *dpp* locus, is still lacking.

4) In paragraph ten, the authors state that "ova and lsd1 mutants exhibited de-repression of a similar subset of transposons" (Figure 3F). However, the analysis on Figure 3 seems to concern germline KDs, and not mutants – please clarify.

Both of them are germline-specific RNAi using UAS-Dcr2; nosGAL4 (referred as GLKD). We have modified our statements as “germline specific knock-down of *ova* and *dlsd1* exhibited de-repression of a similar subset of transposons”.

5) In the Materials and methods section, please verify the source of the ova-RNAi line: it is listed as a BDSC stock, but it seems like it was in fact obtained from the VDRC collection. Please also include the source of the lsd1-RNAi line, as it is not currently listed in the Materials and methods section. Finally, clarify whether or not the UAS-Dcr2 transgene was used in the germline KD analysis.

We have added all the missing information in the Materials and methods section. Note that all the germline-specific RNAi were performed using UAS-Dcr2; nosGAL4 (referred as GLKD throughout the manuscript), and we have updated this information in the manuscript.

6) In subsection “RNA sequencing and computational analysis”, please indicate the manufacture and kits used to generate the small RNA- and RNA-seq libraries.

We used Oligo d(T)25 magnetic beads (NEB, #S1419S) to purify mRNA and NEBNext ultra DNA library prep kits for Illumina (NEB, #E7645S) to generate the library. For small RNA library, we used a VAHTS small RNA library prep kit for Illumina (Vazyme, NR801). We have updated this information in the Materials and methods section.

Reviewer #3:

In Yang et al., the authors generate mutants for ovaries absent (Ova) a previously identified gene in the piRNA pathway and characterize its function. They show (1) Ova is required in the somatic escort cells for germline stem cell differentiation and in the germ line for fertility, (2) Ova acts downstream of Piwi, (3) Interacts with HP1 and dLsd1 regulators of heterochromatin formation and promotes association of HP1a and dLsd1 and (4) Can induce HP1a induced demethylation. I have some major concerns about this paper: (1) The authors switch between germ line function and somatic function without explanation, (2) the phenotypes are not clearly characterized, (3) Some of the controls for the experiments they have carried out are not correct in that they compare ovaries that have late stage egg chambers to ovaries that accumulate undifferentiated cells and (4) changes in ChIPSeq are modest at the best and are not quantitated, and I am not sure are statistically significant.

We thank this reviewer for the thoughtful comments. We will give a brief response here, and detailed responses below in our point-by-point responses. We admit that we mixed descriptions of both germline and somatic functions of *ova* in our study, and this could potentially cause confusion. However, we believe that our phenotype analysis from both somatic and germline cells is a reasonable approach to reveal the basic mechanism of Piwi/piRNA pathway genes. Piwi, the founding member of the Piwi/piRNA pathway, has been demonstrated to have different functions in somatic cells and germline cells. The phenotypes caused by soma and germline-specific depletion of *piwi* are very similar to those caused by the *ova* depletion that we report here. Our mechanistic study reveals that the molecular function of Ova is to link HP1a and dLsd1; consider that the HP1a::dLsd1 fusion protein could rescue both TE and GSC defects. We believe that our study provides a first solid step towards a unified explanation of soma and germline phenotypes caused by the Piwi/piRNA pathway mutations: specifically, Piwi/piRNAs, through recruiting heterochromatic silencing machinery, repress TEs in the germline, and represses regular genes (especially *dpp*) in the somatic escort cells.

Lastly, the model is confusing. The current dogma is that demethylation of H3K4 results loss of K9ac. This then promotes methylation of K9 and which then recruits HP1. In Yang et al's model, how do they propose HP1 can get to its targets in the first place? This needs to clearly explained. I suspect, Ova should affect heterochromatin spreading not formation of heterochromatin. This is something they have not tested nor propose.

We believe what the reviewer is here indicating a general mechanism for heterochromatin establishment (Rudolph et al., 2007). We would note that other studies have shown that bivalent markers (active H3K4me3 and repressive H3K27me3) can be simultaneously present in small genomic regions (e.g., some TEs as in Bernastein *et al.*, (2006); Voigt et al., 2012). In 2015, two studies showed that TEs are co-transcriptionally silenced, and Piwi/piRNA can recruit dLsd1 and Egg for H3K4 demethylation in the promoter region and spread H3K9 methyl downstream of TE body region (Yu et al., 2015; Sienski et al., 2015). In our study, we show that Ova is required for heterochromatic silencing. Meanwhile, *ova* GLKD leads to an increase in the H3K4me2 level at the promoter regions of TEs. In a lacO-GFP reporter assay, we observed that Ova recruitment alone fails to silence the GFP reporter. However, with pre-deposited HP1a to the lacO sites, *ova* overexpression could lead to local H3K4me2 reduction and silencing of the GFP reporter. Results from our co-IP, Y2H, and truncated/ chimeric transgenes experiments suggest that Ova acts as a linker between HP1a and dLsd1. Collectively, our data suggest that Ova acts downstream of HP1a to recruit dLsd1 for local H3K4 demethylation and heterochromatic silencing. This epigenetic connection seems to be evolutionarily conserved, as physical and functional links between H3K4 demethylation and H3K9 methylation have also been reported in yeast (Li et al., 2008).

Major concerns:Figure 1A-D: The authors have used Ova1/4 in all the rescues including figure 1 and 2. They should include the phenotypic characterization of this allelic combination in this figure panel.

Agreed. Consequently, we have now included the *ova[1/4]* phenotypes in Figure 1C, D, G.

Figure 1F-G: This panel needs better pictures as even the "germless" germaria usually stain of anti-spectrin antibody. All the pictures need to be at the same magnification for easy comparison.

We have re-captured the images at the same magnification as in Figure 1C. In the right panel of Figure 1C, the whole germarium is largely germless, except a few leftover germline cells indicated by anti-spectrin and anti-Vasa staining.

Figure 1J-K: The controls should include the non-restrictive temperature and well as C587 at the restrictive temperature at 14 days. 14 days is a long time for restrictive temperature. What happens in 5 -7 days? Also, what happens to protrusions? The idea about protrusions were introduced but then not followed up on.

Thank you for these excellent suggestions. We have included the more appropriate control as well as the image of c587^ts^>ova RNAi at 7 days. As expected, the GSC-like cells gradually accumulate with over time. We did not observe any obvious protrusion phenotype, as GSC-l cells are still wrapped by c587>GFP positive cell processes (Figure1E).

Figure 1M: Again, please include controls for ova4

We have included the ova^4^/+ control in updated Figure 1G.

Figure 1N-S: Please include quantitation for all these phenotypes. As they show a link to piwi and piwi mutant phenotype is quite complicated, they should categorize Ova phenotype in detail.

We have quantified the pMad^+^ and Dad-lacZ^+^ cell number per germarium and included the results in Figure 1I.

Figure 1T: They should not use C587 as a control as this is comparing completely morphologically different ovaries. One enriches for the earlier stages and the other for later stages. They should use bam mutants as a control. Additionally, they should show the RNA seq data and tracks for these experiments for Dpp levels – these data are available to them.

Agreed; we have now removed this piece of qPCR data.

They also report that loss of ova in germ line results in eggs that do not hatch as data not shown. They should report the data.

We have included the fertility test result for *ova* germline mutant females in Figure 1B. The escort cell-specific expression ova in *ova[1/4]* mutant females (referred as *ova* germline mutant) has very limited fertility. The occasional escapers could be due to leaky expression of the *UAS-ova* transgene.

Figure 2: While the results that over expression of ova in piwi mutant ovaries is exciting they should better characterize what aspect of the piwi mutant does this protein rescue? Are the transposons levels in piwi rescued? What about protrusions? What about Dpp levels? What about Ova germ line expression in piwi mutants? Does it rescue loss of fertility in piwi mutants as well? This is important as they switch from somatic effects to germ line effects.

Ova expression in *piwi* mutants (both c587-GAL4 and tub-GAL4) could moderately rescue the oogenesis defects of *piwi* mutants. As shown in the added Figure 2C, escort cell overexpression of *ova* in *piwi* mutant ovaries could bring frequent appearance of developing germline cysts that can be developed into late stage egg chambers. However, the GSC-like tumors still exit (see arrow in Figure 2C). In addition, ubiquitous overexpression of *ova* fails to rescue the TE depression phenotype in *piwi* germline mutants. Therefore, although *ova* appears to be genetically downstream of *piwi*, there must be additional factors downstream of *piwi* that cooperatively function with *ova* to mediate heterochromatic gene silencing.

Figure 3: What happens to heterochromatin levels in ova mutants during oogenesis? Can they stain for H3K9me3 marks?

Good idea. We stained H3K9me3 in control and *ova* GLKD and found no obvious difference (see Author response image 3). Given H3K9me3 is required for both somatic and germline TE silencing and Ova mainly functions to repress germline TE, we speculate Ova is not required for H3K9me3 deposition. However, genome-wide H3K9me3 ChIP-seq experiments will need to be done in the future to confirmatively draw this conclusion.

**Author response image 3. respfig3:** H3K9me3 expression in control and ova GLKD germaria.

Figure 3E: Why are the authors looking for only upregulation of transposons in the germ line when the phenotype is mostly from the soma? They should report transposon levels in somatic loss of function of ova. As the major phenotype of piwi, dlsd1 and ova are from the escort cells. They should carry out an RNAseq for this and report the transposons upregulated in the soma.

We found that somatic loss of *ova* only leads to mild TE upregulation, if any (see Figure 3—figure supplement 3F). Therefore, in somatic escort cells, it appears to us that the most robust change is the transcriptional depression of *dpp*. TEs can also be depressed, but not as robustly. We also used the whole ovary for RNA analysis because of technical feasibility; this could further dampen the depression phenotype, as the most abundant RNAs are from the germline.

Figure 4: In this figure, the authors switch back and forth between the somatic effects and germ line. This is not right. They should be direct that for expediency they use germ line as a read out. They should set up the paper that way as well. The paper reads as if one is discovering the role of Ova in escort cells whereas most of the functional data comes from the germ line. For example 4D vs 4F, in D they are using somatic drivers and in F they use a ubiquitous driver and measure germ line effects.

We admit that we combined both germline and somatic functions of *ova* in our study, and this could potentially cause confusion. However, we believe that phenotype analysis from both somatic and germline cells should be a reasonable approach to reveal the basic mechanism of Piwi/piRNA pathway genes. Similar to other Piwi/piRNA pathway genes, Piwi, the founding member of the Piwi/piRNA pathway, has been demonstrated to have different functions in somatic cells and germline cells. Depletion of *piwi* in somatic escort cells gives rise to GSC-like tumors because of *dpp* derepression; conditional depletion of *piwi* in adult germline causes robust transposon derepression phenotype without alteration of general ovary morphology.

These soma and germline-specific phenotypes are very similar to that caused by *ova* mutation reported here. Our mechanistic study reveals that the molecular function of Ova is to link HP1a and dLsd1, and the HP1a::dLsd1 fusion protein could rescue both TE and GSC defects. We believe that our study provides a first solid step towards a unified explanation of soma and germline phenotypes caused by the Piwi/piRNA pathway mutations: specifically, Piwi/piRNAs, through recruiting heterochromatic silencing machinery, represses TEs in the germline, and represses regular genes (especially *dpp*) in somatic escort cells. Increasing evidence also suggests that the Piwi/piRNA pathway can target many mRNAs in addition to TEs (Shen et al., 2018).

We used different drivers in different cases for specific purposes. In Figure 4, we used the somatic escort cell driver c587 for the purpose of looking at the ovary morphology (germline cyst differentiation) phenotype, as this phenotype is solely caused by the somatic function of *ova*. We used the ubiquitous driver tubGAL4 in cases such as fertility test, because both the somatic and germline functions of *ova* contribute to female fertility.

Figure 4F: Again, using w1118 as control is not right. One does not know if the undifferentiated stages express Het-A and TAHRE at higher levels compared to control. Again, the control and experiment are comparing two different morphologically different ovaries.

As also replied to reviewer 2, we have added more controls, including *ova* RNAi alone, and UAS-Dcr2; nosGAL4 flies, and the TE levels are similar to that in *w1118* flies. Please note that the ova GLKD ovary has normal morphology (similar to the wild type ovary) (Figure 1A, Figure 1—figure supplement 2G), and the GSC/ CB (or refer to as GSC-l) cell number per germarium is also normal (Figure 1—figure supplement 2H). This supports the idea that Ova functions non-autonomously to restrict GSC proliferation. Germline-specific loss of *ova* is sufficient to cause upregulation of Het-A and TAHRE, without changing the ovary morphology.

Figure 5A: Is this germ line or somatic KD? The authors should clearly state this. It is not available in text of figure legends. I assume it is germ line KD. How does this change compare with the genes that are not affected? Is there any quantification for these changes? I am not sure these changes 0-0.75 are real?

It is germline-specific KD (referred as GLKD). We have added the information in the figures and the corresponding figure legends. We have re-analyzed the ChIP-seq data on all the transposons, and H3K4me2 density is significantly increased in Het-A and TAHRE loci but not other loci (see Figure 5—figure supplement 1); We have added the statistical analysis to Figure 5A (see Figure 5B). The Student's *t*-test showed that the observed increases are statistically significant.

There should be quantification for panels A, E and F.

We have included the quantifications as the new Figure 5G.